# Re-Think and Re-Design Graph Neural Networks in Spaces of Continuous Graph Diffusion Functionals

**Tingting Dan**
University of North Carolina at Chapel Hill
tdan@email.unc.edu

**Jiaqi Ding**
University of North Carolina at Chapel Hill
jiaqid@cs.unc.edu

**Ziquan Wei**
University of North Carolina at Chapel Hill
ziquanw@email.unc.edu

**Shahar Z Kovalsky**
University of North Carolina at Chapel Hill
shaharko@unc.edu

**Minjeong Kim**
University of North Carolina at Greensboro
mkim@uncg.edu

**Won Hwa Kim**
Pohang University of Science and Technology
wonhwa@postech.ac.kr

**Guorong Wu**[*]
University of North Carolina at Chapel Hill
grwu@med.unc.edu

## Abstract

Graphs are ubiquitous in various domains, such as social networks and biological systems. Despite the great successes of graph neural networks (GNNs) in modeling and analyzing complex graph data, the inductive bias of locality assumption, which involves exchanging information only within neighboring connected nodes, restricts GNNs in capturing long-range dependencies and global patterns in graphs. Inspired by the classic *Brachistochrone* problem, we seek how to devise a new inductive bias for cutting-edge graph application and present a general framework through the lens of variational analysis. The backbone of our framework is a two-way mapping between the discrete GNN model and continuous diffusion functional, which allows us to design application-specific objective function in the continuous domain and engineer discrete deep model with mathematical guarantees. *First*, we address over-smoothing in current GNNs. Specifically, our inference reveals that the existing layer-by-layer models of graph embedding learning are equivalent to a $\ell_2$-norm integral functional of graph gradients, which is the underlying cause of the over-smoothing problem. Similar to edge-preserving filters in image denoising, we introduce the total variation (TV) to promote alignment of the graph diffusion pattern with the global information present in community topologies. On top of this, we devise a new selective mechanism for inductive bias that can be easily integrated into existing GNNs and effectively address the trade-off between model depth and over-smoothing. *Second*, we devise a novel generative adversarial network (GAN) to predict the spreading flows in the graph through a neural transport equation. To avoid the potential issue of vanishing flows, we tailor the objective function to minimize the transportation within each community while maximizing the inter-community flows. Our new GNN models achieve state-of-the-art (SOTA) performance on graph learning benchmarks such as *Cora*, *Citeseer*, and *Pubmed*.

---

[*]Corresponding author.

37th Conference on Neural Information Processing Systems (NeurIPS 2023).

# 1 Introduction

Graph is a fundamental data structure that arises in various domains, including social network analysis [42], natural language processing [36], computer vision [30], recommender systems [37], and knowledge graphs [21] among others. Tremendous efforts have been made to operate machine learning on graph data (called graph neural networks, or GNNs) at the node [23], link [41], and graph level [27]. The common inductive bias used in GNNs is the homophily assumption that nodes that are connected in a graph are more likely to have similar features or labels. In this context, most GNN models deploy a collection of fully-connected layers to progressively learn graph embeddings by aggregating the nodal feature representations from its topologically-connected neighbors throughout the graph [19].

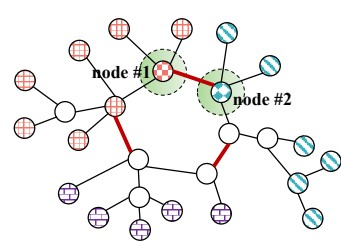

group 1  group 3  group 2  unlabeled

Figure 1: Demonstration of the root cause of over-smoothing in GNNs. Nodes #1 and #2 are located along the boundary of two communities. The locality assumption in GNNs steers the learning of the graph representations by constraining the information exchange via node-to-node connections. However, such link-wise inductive bias opts to neutralize the contrast of node embeddings between nodes #1 and #2, which might undermine the node classification accuracy. Our research framework yields the solution for the over-smoothing issue by enabling heat-kernel diffusion within each community while penalizing the excessive community-to-community information exchanges (highlighted in red).

Under the hood of GNNs, the graph representation learning process is achieved by various learnable operations, such as message passing [5] or graph convolution [23]. Due to the nature of exchanging information in a local graph neighborhood, however, it is challenging to capture global graph representations, which go beyond node-to-node relationship, by leveraging the deep architecture in GNNs while being free of overly smoothing the feature representations for the closely-connected nodes. Fig. 1 demonstrates the root cause of over-smoothing issue in current GNNs, where node color denotes the group label (no color means unlabeled) and edge thickness indicates connection strength. It is clear that nodes #1 and #2 are located at the boundary of two communities. The inductive bias of GNNs (i.e., locality assumption) enforces the node embedding vectors on node #1 and #2 becoming similar due to being strongly connected (highlighted in red), even though the insight of global topology suggests that their node embeddings should be distinct. As additional layers are added to GNNs, the node embeddings become capable of capturing global feature representations that underlie the entire graph topology. However, this comes at the cost of over-smoothing node embeddings across graph nodes due to (1) an increased number of node-to-node information exchanges, and (2) a greater degree of common topology within larger graph neighborhoods. In this regard, current GNNs only deploy a few layers (typically two or three) [26], which might be insufficient to characterize the complex feature representations on the graph.

It is evident that mitigating the over-smoothing problem in GNNs will enable training deeper models. From a network architecture perspective, skip connections [16; 39], residual connections [25; 18], and graph attention mechanisms [34; 33] have been proposed to alleviate the information loss in GNNs, by either preserving the local feature representation or making information exchange adaptive to the importance of nodes in the graph. Although these techniques are effective to patch the over-smoothing issue in some applications, the lack of an in-depth understanding of the root cause of the problem poses the challenge of finding a generalized solution that can be scaled up to current graph learning applications.

Inspired by the success of neural ordinary differential equations in computer vision [10], research focus has recently shifted to link the discrete model in GNNs with partial differential equation (PDE) based numerical recipes [38; 29; 6; 14]. For example, graph neural diffusion (GRAND) formulates GNNs as a continuous diffusion process [6]. In their framework, the layer structure of GNNs corresponds to a specific discretization choice of temporal operators. Since PDE-based model does not revolutionize the underlying inductive bias in current GNNs, it is still unable to prevent the excessive information change between adjacent nodes as in nodes #1 and #2 in Fig. 1. In this regard, using more advanced PDE solvers only can provide marginal improvements in terms of numerical stability over the corresponding discrete GNN models, while the additional computational cost, even in the feed-forward scenario, could limit the practical applicability of PDE-based methods for large-scale graph learning tasks.

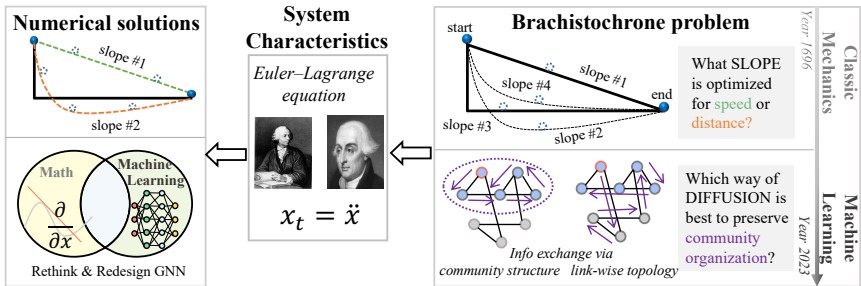

Figure 2: Top: The *Brachistochrone* problem played a pivotal role in the development of classical mechanics and the powerful mathematical tool known as the calculus of variations. Bottom: We introduce a general framework to answer "*Brachistochrone*" problems regarding diffusion patterns on graphs that allow us to re-think and re-design application-specific deep model of GNNs with enhanced mathematical interpretability.

In this regard, pioneering work on continuous approaches has prompted to re-think GNN as a graph diffusion process governed by the Euler-Lagrange (E-L) equation of the heat kernel. This formulation is reminiscent of the *Brachistochrone* problem [2], which emerged over 400 years ago and established the mathematical framework of classical mechanics. The powerful calculus of variations allows us to generate solutions for various mechanics questions (e.g., the slope that yields the fastest ball sliding down the curve is given by a cycloid) through the lens of E-L equation, as shown in Fig. 2 (top).

In a similar vein, the question that arises in the context of community detection is: What graph diffusion pattern is best suited for preserving community organizations? The question for graph classification would be: What graph diffusion pattern works best for capturing the system-level characteristics of graph topology? Following the spirit of *Brachistochrone* problem, we present a general research framework to customize application-specific GNNs in a continuous space of graph diffusion functionals. As shown in Fig. 2 (bottom), we have established a fundamental structure for our framework that involves a two-way mapping between a discrete GNN model and a continuous graph diffusion functional. This allows us to develop application-specific objective functions (with an explainable regularization term) in the continuous domain and construct a discrete deep model with mathematical guarantee. We demonstrate two novel GNN models, one for addressing over-smoothing and one for predicting the flows from longitudinal nodal features, both achieving state-of-the-art performance (*Cora*: 85.6%, *Citeseer*: 73.9%, *Pubmed*: 80.10%, even in 128 network layers).

We have made four major contributions. (1) We establish a connection between the discrete model of GNNs and the continuous functional of inductive bias in graph learning by using the E-L equation as a stepping stone to bridge the discrete and continuous domains. (2) We introduce a general framework to re-think and re-design new GNNs that is less "black-box". (3) We devise a novel selective mechanism upon inductive bias to address the over-smoothing issue in current GNNs and achieve state-of-the-art performance on graph learning benchmarks. (4) We construct a novel GNN in the form of a generative adversarial network (GAN) to predict the flow dynamics in the graph by a neural transport equation.

## 2 Methods

In the following, we first elucidate the relationship between GNN, PDE, and calculus of variations (COV), which sets the stage for the *GNN-PDE-COV* framework for new GNN models in Section 2.2.

### 2.1 Re-think GNNs: Connecting dots across graph neural networks, graph diffusion process, Euler-Lagrange equation, and Lagrangian mechanics

**Graph diffusion process.** Given graph data $\mathcal{G} = (V, W)$ with $N$ nodes $V = \{v_i | i = 1, \ldots, N\}$, the adjacency matrix $W = [w_{ij}]_{i,j=1}^{N} \in \mathbb{R}^{N \times N}$ describes connectivity strength between any two nodes. For each node $v_i$, we have a graph embedding vector $x_i \in \mathbb{R}^m$. In the context of graph topology, the graph gradient $(\nabla_{\mathcal{G}} x)_{ij} = w_{ij} (x_i - x_j)$ indicates the feature difference between $v_i$ and $v_j$ weighted

---

[2]The *Brachistochrone* problem is a classic physics problem that involves finding the curve down which a bead sliding under the influence of gravity will travel in the least amount of time between two points.

by the connectivity strength $w_{ij}$, where $\nabla_{\mathcal{G}}$ is a $\mathbb{R}^N \to \mathbb{R}^{N \times N}$ operator. Thus, the graph diffusion process can be formulated as $\frac{\partial x(t)}{\partial t} = div\left(\nabla_{\mathcal{G}} x(t)\right)$, where the evolution of embedding vectors $x = [x_i]_{i=1}^N$ is steered by the graph divergence operator.

**Connecting GNN to graph diffusion.** In the regime of GNN, the regularization in the loss function often measures the smoothness of embeddings $x$ over the graph by $x^T \Delta x$, where $\Delta = div(\nabla_{\mathcal{G}})$ is the graph Laplacian operator [23]. To that end, the graph smoothness penalty encourages two connected nodes to have similar embeddings by information exchange in each GNN layer. Specifically, the new graph embedding $x^l$ in the $l^{th}$ layer is essentially a linear combination of the graph embedding $x^{l-1}$ in the previous layer, i.e., $x^l = A_{W,\Theta} x^{l-1}$, where the matrix $A$ depends on graph adjacency matrix $W$ and trainable GNN parameter $\Theta$. After rewriting $x^l = Ax^{l-1}$ into $x^l - x^{l-1} = (A - I)x^{l-1}$, updating graph embeddings $x$ in GNN falls into a discrete graph diffusion process, where the time parameter $t$ acts as a continuous analog of the layers in the spirit of Neural ODEs [10]. It has been shown in [6] that running the graph diffusion process for multiple iterations is equivalent to applying a GNN layer multiple times.

**GNN is a discrete model of Lagrangian mechanics via E-L equation.** The diffusion process $\frac{\partial x(t)}{\partial t} = div\left(\nabla_{\mathcal{G}} x(t)\right)$ has been heavily studied in image processing for decades ago, which is the E-L equation of the functional $\min_x \int_\Omega |\nabla x|^2 dx$. By replacing the 1D gradient operator defined in the Euclidean space $\Omega$ with the graph gradient $(\nabla_{\mathcal{G}} x)_{ij}$, it is straightforward to find that the equation governing the graph diffusion process $\frac{\partial x(t)}{\partial t} = div\left(\nabla_{\mathcal{G}} x(t)\right)$ is the E-L equation of the functional $\min_x \int_{\mathcal{G}} |\nabla_{\mathcal{G}} x|^2 dx$ over the graph topology. Since the heat kernel diffusion is essentially the mathematical description of the inductive bias in current GNNs [6; 14], we have established a mapping between the mechanics of GNN models and the functional of graph diffusion patterns in a continuous domain.

**Tracing the smoking gun of over-smoothing in GNNs.** In Fig. 1, we observed that the inductive bias of link-wise propagation is the major reason for excessive information exchange, which is attributed to the over-smoothing problem in GNNs. An intuitive approach is to align the diffusion process with high-level properties associated with graph topology, such as network communities. After connecting the GNN inductive bias to the functional of graph diffusion process, we postulate that the root cause of over-smoothing is the isotropic regularization mechanism encoded by the $\ell_2$-norm. More importantly, connecting GNN to the calculus of variations offers a more principled way to design new deep models with mathematics guarantees and model mechanistic explainability.

## 2.2 Re-design GNNs: Revolutionize inductive bias, derive new E-L equation, and construct deeper GNN

The general roadmap for re-designing GNNs typically involves three major steps: (1) formulating inductive bias into the functional of graph diffusion patterns; (2) deriving the corresponding E-L equation; and then (3) developing a new deep model of GNN based on the finite difference solution of E-L equation. Since the graph diffusion functional is application-specific, we demonstrate the construction of new GNN models in the following two graph learning applications.

### 2.2.1 Develop *VERY* deep GNNs with a selective mechanism for link-adaptive inductive bias

**Problem formulation.** Taking the feature learning component (learnable parameters $\Theta$) out of GNNs, the graph embeddings $x^L$ (output of an $L$-layer GNN) can be regarded as the output of an iterative smoothing process ($L$ times) underlying the graph topology $\mathcal{G}$, constrained by the data fidelity $\left\| x^L - x^0 \right\|_2^2$ (w.r.t. the initial graph embeddings $x^0$) and graph smoothness term $\int_{\mathcal{G}} |\nabla_{\mathcal{G}} x|^2 dx$. Inspired by the great success of total variation (TV) for preserving edges in image denoising [31], reconstruction [35] and restoration [8], we proposed to address the over-smoothing issue in current GNN by replacing the quadratic Laplacian regularizer with TV on graph gradients, i.e., $\mathcal{J}_{TV}(x) = \int |\nabla_{\mathcal{G}} x| dx$. Thus, the TV-based objective function for graph diffusion becomes: $\min_x (\left\| x - x^0 \right\|_2^2 + \mathcal{J}_{TV}(x))$.

However, the $\ell_1$-norm function, denoted by $|\cdot|$ in the definition of the total variation functional $\mathcal{J}_{TV}$, is not differentiable at zero. Following the dual-optimization schema [4; 7], we introduce the latent

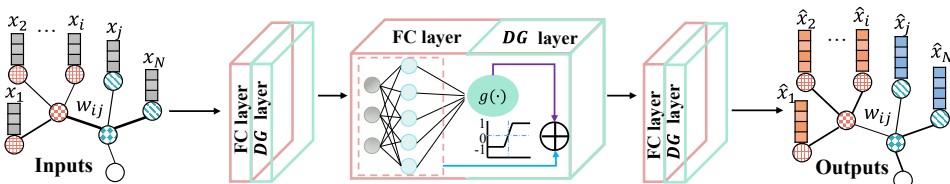

Figure 3: Our new deep model integrates a novel diffusion-gathering (*DG*) layer (for selective graph diffusion) after the conventional fully-connected (FC) layer (for graph representation learning).

auxiliary matrix $z \in \mathbb{R}^{N \times N}$ and reformulate the TV-based functional as $\min_{x} \max_{z} \mathcal{J}_{TV}(x, z) = \max_{z} \min_{x} \int (z \otimes \nabla_{\mathcal{G}} x) dx$, subject to $|z| \leq \mathbf{1}^{N \times N}$, where $\otimes$ is Hadamard operation between two matrices. Furthermore, we use an engineering trick of element-wise operation $z_{ij}(\nabla_{\mathcal{G}} x)_{ij}$ to keep the degree always non-negative (same as we take the absolute value), which makes the problem solvable. In the end, we reformulate the minimization of $\mathcal{J}_{TV}(x)$ into a dual *min-max* functional as $\mathcal{J}_{TV}(x, z)$, where we maximize $z$ ($z \to \mathbf{1}^{N \times N}$) such that $\mathcal{J}_{TV}(x, z)$ is close enough to $\mathcal{J}_{TV}(x)$. Therefore, the new objective function is reformulated as:

$$\mathcal{J}(x, z) = \max_{z} \min_{x} \left\| x - x^0 \right\|_2^2 + \lambda \int (z \nabla_{\mathcal{G}} x) dx, \tag{1}$$

where $\lambda$ is a scalar balancing the data fidelity term and regularization term. Essentially, Eq. 1 is the dual formulation with *min-max* property for the TV distillation problem [44].

**Constructing E-L equations.** To solve Eq. 1, we present the following two-step alternating optimization schema. *First*, the inner minimization problem (solving for $x_i$) in Eq. 1 can be solved by letting $\frac{\partial}{\partial x_i} \mathcal{J}(x_i, z_i) = 0$:

$$\frac{\partial}{\partial x_i} \mathcal{J}(x_i, z_i) = 2(x_i - x_i^0) + \lambda z_i \nabla_{\mathcal{G}} x_i = 0 \quad \Rightarrow \quad \hat{x}_i = x_i^0 - \frac{\lambda}{2} z_i \nabla_{\mathcal{G}} x_i \tag{2}$$

Replacing $(\nabla_{\mathcal{G}} x)_{ij}$ with $w_{ij}(x_i - x_j)$, the intuition of Eq. 2 is that each element in $\hat{x}_i$ is essentially the combination between the corresponding initial value in $x_i^0$ and the overall graph gradients $z_i \nabla_{\mathcal{G}} x_i = \sum_{j \in \mathcal{N}_i} w_{ij}(x_i - x_j) z_i$ within its graph neighborhood $\mathcal{N}_i$. In this regard, Eq. 2 characterizes the dynamic information exchange on the graph, which is not only steered by graph topology but also moderated by the attenuation factor $z_i$ at each node.

*Second*, by substituting Eq. 2 back into Eq. 1, the objective function of $z_i$ becomes $\mathcal{J}(z_i) = \max_{|z_i| \leq \mathbf{1}} \left\| \frac{\lambda}{2} z_i \nabla_{\mathcal{G}} x_i \right\|_2^2 + \lambda z_i \nabla_{\mathcal{G}} (x_i^0 - \frac{\lambda}{2} z_i \nabla_{\mathcal{G}} x_i)$. With simplification (in Eq. S1 to Eq. S3 of Supplementary), the optimization of each $z_i$ is achieved by $\arg \min_{|z_i| \leq \mathbf{1}} z_i \nabla_{\mathcal{G}} x_i z_i \nabla_{\mathcal{G}} x_i - \frac{4}{\lambda} z_i \nabla_{\mathcal{G}} x_i^0$.

Specifically, we employ the majorization-minimization (MM) method [15] to optimize $z_i$ by solving this constrained minimization problem (the detailed derivation process is given in Eq. S4 to S12 of Section S1.1 of Supplementary), where $z_i$ can be iteratively refined by:

$$z_i^l = g(\underbrace{z_i^{l-1} + \frac{2}{\beta \lambda} \nabla_{\mathcal{G}} x_i}_{b}, 1) = \begin{cases} b & |b| \leq 1 \\ 1 & b > 1 \\ -1 & b < -1 \end{cases} \tag{3}$$

$\beta$ is a hyper-parameter that is required to be no less than the largest eigenvalue of $(\nabla_{\mathcal{G}} x_i)(\nabla_{\mathcal{G}} x_i)^{\intercal}$.

**Develop new GNN network architecture with a selective inductive bias.** The building block in vanilla GNN [23] is a FC (fully-connected) layer where the input is the embedding vectors after isotropic graph diffusion (in $\ell_2$-norm). Since the estimation of graph embeddings $x$ in Eq. 2 depends on the latest estimation of $z^{(l)}$, such recursive *min-max* solution for Eq. 1 allows us to devise a new network architecture that disentangles the building block in vanilla GNN into the feature representation learning and graph diffusion underlying TV. As shown in Fig. 3, we first deploy a FC layer to update the graph embeddings $x^{(l)}$. After that, we concatenate a diffusion-gathering (*DG*) layer for selective graph diffusion, which sequentially applies (1) node-adaptive graph diffusion (blue

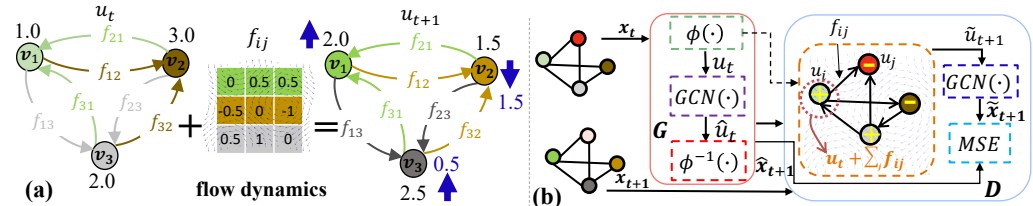

Figure 4: (a) The illustration of the computational challenge for estimating the spreading flow. (b) The GAN architecture for *min-max* optimization in the network.

arrow in Fig. 3) on $x^{(l)}$ by Eq. 2 [3], and (2) gathering operation $g(\cdot)$ (purple arrow in Fig. 3) to each $x_i^{(l)}$ by Eq. 3.

**Remarks.** Eq. 3 indicates that larger connective degree results in larger value of $z$. Thus, the *DG* layer shifts the diffusion patterns by penalizing the inter-community information exchange (due to strong connections) while remaining the heat-kernel diffusion within the community. The preference of such link-adaptive diffusion can be adjusted by the hyper-parameter $\lambda$ [4] in Eq. 1. Recall our intuitive solution for over-smoothing problem in Fig. 1, the *DG* layer offers the exact global insight of graph topology to keep the node embeddings distinct between nodes #1 and #2. We demonstrate the effect of *DG* layer on the real-world graph data in Fig. S3 of Supplementary document.

### 2.2.2 Predict flow dynamics through graph neural transport equation

**Problem formulation.** We live in a world of complex systems, where everything is intricately connected in multiple ways. A holistic insight of how the system's components interact with each other and how changes in one part of the system can affect the behavior of the whole sheds new light on the dynamic behaviors of these complex systems over time. However, oftentimes it is an ill-posed problem. Taking the toy system in Fig. 4(a) as an example, while it is simple to calculate the future focal patterns based on the focal patterns at the current time point and the node-to-node flow information, determining flow dynamics based on longitudinal nodal observations is computationally hard since the solution is not unique.

The naïve solution to predict the spreading flow is to (1) train a GNN to learn the intrinsic node embeddings and (2) predict the flow based on the difference of learned embeddings. However, this two-step approach might suffer from vanishing flow due to over-smoothing in GNNs. Following the spirit of *Brachistochrone* problem, we ask the question "What flow field $f(t) = [f_{ij}(t)]_{i,j=1}^N$ underlines the system mechanics to the extent that it is able to predict the behaviors in the future?"

In this paper, we focus on the conservative system of energy transportation [2]. The system mechanics is formulated as:

$$\frac{dx}{dt} + div(q) = 0 \tag{4}$$

where $q = [q_{ij}]_{i,j=1}^N$ is the flux field which propagates the potential energy $u(t) = [u_i(t)]_{i=1}^N$ (conserved quantity) over time. Similar to a gravity field driving water flow, the intuition of Eq. 4 is that the change of energy density $u$ (we assume there is a non-linear mapping $\phi$ from external force $x$ to $u$, i.e., $u_i = \phi(x_i)$) leads to energy transport throughout the entire graph. As flux is closely related to the difference of energy $\nabla_{\mathcal{G}} u$ underlying the graph topology, we assume the energy flux $q$ is regulated by the potential energy field $\nabla_{\mathcal{G}} u$, i.e., $q = \alpha \otimes \nabla_{\mathcal{G}} u$, where $\alpha = [\alpha_{ij}]_{i,j=1}^N$ is a learnable matrix characterizing the link-wise contribution of each energy potential $\nabla_{\mathcal{G}} u_{ij}$ to the potential energy flux $q_{ij}$.

By plugging $q = \alpha \otimes \nabla_{\mathcal{G}} u$ into Eq. 4, the energy transport process can be reformulated as:

$$\frac{\partial u}{\partial t} = -\phi^{-1}(\alpha \otimes div(\nabla_{\mathcal{G}} u)) = -\phi^{-1}(\alpha \otimes \Delta u), \tag{5}$$

where $\Delta = div(\nabla_{\mathcal{G}})$ is the graph Laplacian operator. Since the PDE in Eq. 5 is equivalent to the E-L equation of the quadratic functional $\mathcal{J}(u) = \min_u \int_{\mathcal{G}} \alpha \otimes |\nabla_{\mathcal{G}} u|^2 du$ (after taking $\phi$

---

[3] Since the optimization schema has been switched to the layer-by-layer manner, the initialization $x_0$ becomes $x^{(l-1)}$ from the previous layer.

[4] $\lambda$ can be either pre-defined or learned from the data.

away), a major issue is the over-smoothness in $u$ that might result in vanishing flows. In this context, we propose to replace the $\ell_2$-norm integral functional $\mathcal{J}(u)$ with TV-based counterpart $\mathcal{J}_{TV}(u) = \min_u \int_\mathcal{G} \alpha \otimes |\nabla_\mathcal{G} u| du$.

**Renovate new E-L equation – graph neural transport equations.** Following the *min-max* optimization schema in Eq. 1-3, we introduce an auxiliary matrix $f$ to lift the undifferential-able barrier. After that, the minimization of $\mathcal{J}_{TV}(u)$ boils down into a dual *min-max* functional $\mathcal{J}_{TV}(u, f) = \min_u \max_f \int_\mathcal{G} \alpha \otimes f(\nabla_\mathcal{G} u) du$. Since $u(t)$ is a time series, it is difficult to derive the deterministic solutions (as Eq. 2-3) by letting $\frac{\partial}{\partial u} \mathcal{J}_{TV} = 0$ and $\frac{\partial}{\partial f} \mathcal{J}_{TV} = 0$. Instead, we use *Gâteaux* variations to optimize $\mathcal{J}_{TV}(u, f)$ via the following two coupled time-dependent PDEs (please see Section S1.2, Eq. S14 to Eq. S19, in Supplementary for details):

$$
\begin{cases}
\max_f \frac{df}{dt} = \alpha \otimes \nabla_\mathcal{G} u \\
\min_u \frac{du}{dt} = \alpha \otimes div(f)
\end{cases}
\tag{6}
$$

**Remarks.** The solution to Eq. 6 is known as continuous max-flow and constitutes a continuous version of a graph-cut [1]. Since $\alpha$ is a latent variable and potential energy $u$ is given, the maximization of $f$ opts towards maximizing the spreading flow through the lens of $\alpha$. In this regard, the mechanistic role of auxiliary matrix $f$ is essentially the latent (maximized) spreading flows that satisfy $u(t+1)_i = u(t)_i + \sum_{j=1}^N f_{ij}(t)$. The potential energy $\hat{u}$ can be solved via a wave equation ($u_{tt} = div(f_t) = \alpha^2 \otimes \Delta u$), where the system dynamics is predominated by the adjusted Lagrangian mechanics $\alpha^2 \otimes \Delta u$. By determining $\alpha$ at a granularity of graph links, we devise a novel GAN model to predict the spreading flows $f$ which not only offers explainability underlying the *min-max* optimization mechanism in Eq. 6 but also sets the stage to understand system dynamics through machine learning.

**Develop a GAN model of flow prediction with TV-based Lagrangian Mechanics.** The overall network architecture is shown in Fig. 4 (b), which consists of a generator (red solid box) and a discriminator module (blue solid box). Specifically, the generator ($G$) consists of (1) a GCN component [14] to optimize $\hat{u}$ through the wave equation $u_{tt} = \alpha^2 \otimes \Delta u$ and (2) a FC layer to characterize the non-linear mapping function $\hat{x}(t+1) = \phi^{-1}(\hat{u}(t))$. In contrast, the discriminator ($D$) is designed to (1) synthesize $\alpha$ and (2) construct the future $\tilde{u}_{t+1}$ based on the current $u_t$ and current estimation of spreading flow $f = \alpha \otimes \nabla_\mathcal{G} u$ (orange dash box). To make the network architecture consistent between generator and discriminator modules, we include another GCN to map the synthesized $\tilde{u}(t+1)$ to the external force $\tilde{x}(t+1)$. Note, since the working mechanism of this adversarial model underlines the min-max optimization in the energy transport equation, the nature of predicted spreading flows is carved by the characteristics of *max-flow*.

The driving force of our network is to minimize (1) the MSE (mean square error) between the output of the generator $\hat{x}_{t+1}$ and the observed regional features, (2) the distance between the synthesized regional features $\tilde{x}_{t+1}$ (from the discriminator) and the output of generator $\hat{x}_{t+1}$ (from the generator). In the spirit of probabilistic GAN [43], we use one loss function $\mathcal{L}_D$ to train the discriminator ($D$) and another one $\mathcal{L}_G$ to train the generator ($G$):

$$
\begin{cases}
\mathcal{L}_D = D\left(x_{t+1}\right) + [\xi - D\left(G\left(x_t\right)\right)]^+ \\
\mathcal{L}_G = D\left(G\left(x_t\right)\right)
\end{cases}
\tag{7}
$$

where $\xi$ denotes the positive margin and the operator $[\cdot]^+ = \max(0, \cdot)$. Minimizing $\mathcal{L}_G$ is similar to maximizing the second term of $\mathcal{L}_D$ except for the non-zero gradient when $D(G(x_t)) \geq \xi$.

## 3 Experiments

In this section, we evaluate the performance of the proposed *GNN-PDE-COV* framework with comparison to six graph learning benchmark methods on a wide variety of open graph datasets [32], as well as a proof-of-concept application of uncovering the propagation pathway of pathological events in Alzheimer's disease (AD) from the longitudinal neuroimages.

### 3.1 Datasets and experimental setup

**Dataset and benchmark methods.** We evaluate the new GNN models derived from our proposed GNN framework in two different applications. *First*, we use three standard citation networks, namely

*Cora*, *Citeseer*, and *Pubmed* [32] for node classification (the detailed data statistic is shown in Table S2 of Supplementary). We adopt the public fixed split [40] to separate these datasets into training, validation, and test sets. We follow the experimental setup of [9] for a fair comparison with six benchmark GNN models (vanilla GCN [23], GAT [34], GCNII [9], ResGCN [25], DenseGCN [25], GRAND [6]). Since our *DG*-layer can be seamlessly integrated into existing GNNs as a plug-in. The corresponding new GNN models (with *DG*-layer) are denoted GCN+, GAT+, GCNII+, ResGCN+, DenseGCN+, and GRAND+, respectively.

*Second*, we apply the GAN model in Section 2.2.2 to predict the concentration level of AD-related pathological burdens and their spreading pathways from longitudinal neuroimages. Currently, there is no *in-vivo* imaging technique that can directly measure the flow of information across brain regions. Here, our computational approach holds great clinical value in understanding the pathophysiological mechanism involved in disease progression [22]. Specifically, we parcellate each brain into 148 cortical surface regions and 12 sub-cortical regions using Destrieux atlas [12]. The wiring topology of these 160 brain regions is measured by diffusion-weighted imaging [3] and tractography techniques [17]. The regional concentration levels AD pathology including amyloid, tau, and fluorodeoxyglucose (FDG) and cortical thickness (CoTh) are measured from PET (positron emission tomography) and MRI (magnetic resonance imaging) scans [20]. We use a total of $M = 1,291$ subjects from ADNI [28], each having longitudinal imaging data (2-5 time points). The details of image statistics and pre-processing are shown in Sec. S2.1.2. Since we apply the flow prediction model to each modality separately, we differentiate them with *X-FlowNet* ($X$ stands for amyloid, tau, FGD, and CoTh).

**Experimental setup**. In the node classification task, we verify the effectiveness and generality of *DG*-layer in various number of layers ($L = 2, 4, 8, 16, 32, 64, 128$). All baselines use their default parameter settings. Evaluation metrics include accuracy, precision and F1-score. To validate the performance of *X-FlowNet*, we examine (1) prediction accuracy (MAE) of follow-up concentration level, (2) prediction of the risk of developing AD using the baseline scan, and (3) understand the propagation mechanism in AD by revealing the node-to-node spreading flows of neuropathologies.

The main results of graph node classification and flow prediction are demonstrated in Section 3.2 and 3.3, respectively. Other supporting results such as ablation study and hyper-parameter setting are shown in Section S2 of the Supplementary document.

## 3.2 Experimental results on graph node classification

We postulate that by mitigating the over-smoothing issue, we can leverage the depth of GNN models to effectively capture complex feature representations in graph data. As shown in Table 1, we investigate the graph node classification accuracy as we increase the number of GNN layers by six benchmark GNN models and their corresponding plug-in models (indicated by '+' at the end of each GNN model name) with the *DG*-layer. The results demonstrate that: (1) the new GNN models generated from the *GNN-PDE-COV* framework have achieved SOTA in *Cora* (86.30% by GCNII+), *Citeseer* (75.65% by GRAND+), and *Pubmed* (80.10 % by GCNII+); (2) all of new GNN models outperforms their original counterparts with significant improvement in accuracy; (3) the new GNN models exhibit less sensitivity to the increase of model depth compared to current GNN models; (4) the new GNN models are also effective in resolving the gradient explosion problem [26] (e.g, the gradient explosion occurs when training GAT on all involved datasets with deeper than 16 layers, while our GAT+ can maintain reasonable learning performance even reaching 128 layers.)

It is important to note that due to the nature of the graph diffusion process, graph embeddings from all GNN models (including ours) will eventually become identical after a sufficiently large number of layers [11]. However, the selective diffusion mechanism (i.e., penalizing excessive diffusion across communities) provided by our *GNN-PDE-COV* framework allows us to control the diffusion patterns and optimize them for specific graph learning applications.

## 3.3 Application for uncovering the propagation mechanism of pathological events in AD

*First*, we evaluate the prediction accuracy between the ground truth and the estimated concentration values by our *X-FlowNet* and six benchmark GNN methods. The statistics of MAE (mean absolute error) by *X-FlowNet*, GCN, GAT, GRAND, ResGCN, DenseGCN and GCNII, at different noise levels on the observed concentration levels, are shown in Fig. 5 (a). It is clear that our *X-FlowNet* consistently outperforms the other GCN-based models in all imaging modalities.

Table 1: Test accuracies (%) on citation networks. We show the mean value, the quota of increase (↑)/decrease(↓) after adding *DG* layer. Statistical significance is determined from 50 resampling tests. '∗' means statistically significance with $p \leq 0.05$, '∗∗' denotes $p \leq 0.01$. The missing results are due to the huge consumption of GPU memory for large graphs (DenseGCN) or gradient explosions (GAT) or non-convergence (GRAND). The best performance of baselines is denoted in blue, while the best performance after adding the *DG* layer is denoted in red.

| Dataset | Model | $L=2$ | $L=4$ | $L=8$ | $L=16$ | $L=32$ | $L=64$ | $L=128$ |
|---|---|---|---|---|---|---|---|---|
| **Cora** | GCN | 81.30 | 79.90 | 75.70 | 25.20 | 20.00 | 31.80 | 20.80 |
| | **GCN+** | $82.70^{**}_{1.40\uparrow}$ | $82.70^{**}_{2.80\uparrow}$ | $82.30^{**}_{6.60\uparrow}$ | $70.60^{**}_{45.4\uparrow}$ | $67.80^{**}_{47.8\uparrow}$ | $66.60^{**}_{34.8\uparrow}$ | $59.90^{**}_{39.1\uparrow}$ |
| | GAT | 82.40 | 80.30 | 57.90 | 31.90 | – | – | – |
| | **GAT+** | $82.60^{**}_{0.20\uparrow}$ | $80.50^{**}_{0.20\uparrow}$ | $69.70^{**}_{11.8\uparrow}$ | $66.00^{**}_{34.1\uparrow}$ | $63.60^{**}_{63.6\uparrow}$ | $54.60^{**}_{54.6\uparrow}$ | $45.70^{**}_{45.7\uparrow}$ |
| | GRAND | 80.00 | 82.64 | 82.74 | 83.45 | 81.83 | 80.81 | 79.19 |
| | **GRAND+** | $81.93^{**}_{1.93\uparrow}$ | $83.45^{**}_{0.81\uparrow}$ | $82.95^{**}_{0.20\uparrow}$ | $84.27^{**}_{1.32\uparrow}$ | $83.15^{**}_{0.71\uparrow}$ | $81.52^{**}_{0.71\uparrow}$ | $80.10^{**}_{0.91\uparrow}$ |
| | ResGCN | 76.30 | 77.30 | 76.20 | 77.60 | 73.30 | 31.90 | 31.90 |
| | **ResGCN+** | $77.80^{**}_{1.50\uparrow}$ | $78.70^{**}_{1.40\uparrow}$ | $78.80^{**}_{2.60\uparrow}$ | $78.60^{**}_{1.00\uparrow}$ | $76.90^{**}_{3.60\uparrow}$ | $76.80^{**}_{44.9\uparrow}$ | $33.60^{**}_{1.70\uparrow}$ |
| | DenseGCN | 76.60 | 78.50 | 76.00 | – | – | – | – |
| | **DenseGCN+** | $78.00^{**}_{1.40\uparrow}$ | $78.70^{**}_{0.20\uparrow}$ | $76.90^{**}_{1.40\uparrow}$ | – | – | – | – |
| | GCNII | 76.40 | 81.90 | 81.50 | 84.80 | 84.60 | 85.50 | 85.30 |
| | **GCNII+** | $84.70^{**}_{8.30\uparrow}$ | $84.80^{**}_{2.90\uparrow}$ | $84.70^{**}_{3.20\uparrow}$ | $85.20^{**}_{0.40\uparrow}$ | $85.40^{**}_{0.80\uparrow}$ | $86.30^{**}_{0.80\uparrow}$ | $85.60_{0.30\uparrow}$ |
| **Citeseer** | GCN | 70.20 | 62.50 | 62.90 | 21.00 | 17.90 | 22.90 | 19.80 |
| | **GCN+** | $72.90^{**}_{2.70\uparrow}$ | $67.30^{**}_{4.80\uparrow}$ | $72.00^{**}_{9.10\uparrow}$ | $54.70^{**}_{33.7\uparrow}$ | $50.30^{**}_{32.4\uparrow}$ | $48.40^{**}_{25.5\uparrow}$ | $46.60^{**}_{26.8\uparrow}$ |
| | GAT | 71.70 | 58.60 | 26.60 | 18.10 | – | – | – |
| | **GAT+** | $73.00^{**}_{1.30\uparrow}$ | $69.50^{**}_{10.9\uparrow}$ | $47.60^{**}_{21.0\uparrow}$ | $31.80^{**}_{13.7\uparrow}$ | $31.30^{**}_{31.3\uparrow}$ | $30.60^{**}_{30.6\uparrow}$ | $29.30^{**}_{29.3\uparrow}$ |
| | GRAND | 71.94 | 72.58 | 73.87 | 75.00 | 75.16 | 72.90 | 69.52 |
| | **GRAND+** | $72.26^{*}_{0.32\uparrow}$ | $73.55^{**}_{0.97\uparrow}$ | $75.16^{**}_{1.29\uparrow}$ | $75.65^{**}_{0.65\uparrow}$ | $75.52^{**}_{0.36\uparrow}$ | $74.52^{*}_{1.62\uparrow}$ | $72.26^{**}_{2.74\uparrow}$ |
| | ResGCN | 67.10 | 66.00 | 63.60 | 65.50 | 62.3 | 18.80 | 18.10 |
| | **ResGCN+** | $68.00^{**}_{0.90\uparrow}$ | $67.60^{**}_{1.60\uparrow}$ | $66.00^{**}_{2.40\uparrow}$ | $66.00^{**}_{0.50\uparrow}$ | $65.80^{**}_{3.50\uparrow}$ | $24.00^{**}_{5.20\uparrow}$ | $24.30^{**}_{6.20\uparrow}$ |
| | DenseGCN | 67.40 | 64.00 | 62.20 | – | – | – | – |
| | **DenseGCN+** | $67.80^{*}_{0.40\uparrow}$ | $66.60^{**}_{2.60\uparrow}$ | $64.70^{**}_{2.50\uparrow}$ | – | – | – | – |
| | GCNII | 66.50 | 67.80 | 69.30 | 71.60 | 73.10 | 71.40 | 70.20 |
| | **GCNII+** | $72.40^{**}_{5.90\uparrow}$ | $73.30^{**}_{5.5\uparrow}$ | $73.80^{**}_{4.50\uparrow}$ | $73.40^{**}_{1.80\uparrow}$ | $73.80^{**}_{0.70\uparrow}$ | $74.60^{**}_{3.20\uparrow}$ | $73.90^{**}_{3.70\uparrow}$ |
| **Pubmed** | GCN | 78.50 | 76.50 | 77.30 | 40.90 | 38.20 | 38.10 | 38.70 |
| | **GCN+** | $79.80^{**}_{1.30\uparrow}$ | $79.10^{**}_{2.60\uparrow}$ | $78.20^{**}_{0.90\uparrow}$ | $77.40^{**}_{36.5\uparrow}$ | $76.20^{**}_{38.0\uparrow}$ | $75.10^{**}_{37.0\uparrow}$ | $73.00^{**}_{34.3\uparrow}$ |
| | GAT | 77.40 | 72.20 | 77.80 | 40.70 | – | – | – |
| | **GAT+** | $77.90^{*}_{0.50\uparrow}$ | $77.30^{**}_{5.10\uparrow}$ | $78.50^{*}_{0.70\uparrow}$ | $73.50^{**}_{32.8\uparrow}$ | $68.20^{**}_{68.2\uparrow}$ | $66.80^{**}_{66.8\uparrow}$ | $63.50^{**}_{63.5\uparrow}$ |
| | GRAND | 77.07 | 77.94 | 78.29 | 79.93 | 79.12 | – | – |
| | **GRAND+** | $78.03^{**}_{0.96\uparrow}$ | $78.34^{*}_{0.40\uparrow}$ | $80.21^{**}_{1.92\uparrow}$ | $80.08^{**}_{0.15\uparrow}$ | $79.32_{0.20\uparrow}$ | – | – |
| | ResGCN | 76.30 | 77.30 | 76.20 | 77.60 | 73.30 | 31.90 | 31.90 |
| | **ResGCN+** | $77.80^{**}_{1.50\uparrow}$ | $78.70^{**}_{1.40\uparrow}$ | $78.80^{*}_{2.60\uparrow}$ | $78.60^{*}_{1.00\uparrow}$ | $76.90^{**}_{3.60\uparrow}$ | $76.80^{**}_{44.90\uparrow}$ | $32.00_{0.10\uparrow}$ |
| | DenseGCN | 75.80 | 76.10 | 75.80 | – | – | – | – |
| | **DenseGCN+** | $76.10_{0.30\uparrow}$ | $76.70^{*}_{0.60\uparrow}$ | $77.50^{**}_{1.70\uparrow}$ | – | – | – | – |
| | GCNII | 77.30 | 78.80 | 79.50 | 79.70 | 79.90 | 0.7980 | 79.70 |
| | **GCNII+** | $78.40^{**}_{1.10\uparrow}$ | $80.10^{**}_{1.30\uparrow}$ | $80.00^{*}_{0.60\uparrow}$ | $80.10^{**}_{0.30\uparrow}$ | $80.00_{0.20\uparrow}$ | $80.00_{0.20\uparrow}$ | $80.10^{*}_{0.40\uparrow}$ |

*Second*, we have evaluated the potential of disease risk prediction, which can be regarded as a graph classification problem. We assume that we have baseline amyloid, tau, FDG, and CoTh scans, and evaluate the prediction accuracy, precision and F1-score of various models in forecasting the risk of developing AD. We consider two dichotomous cases: one included only AD vs. CN groups and the other involved AD/LMCI vs. CN/EMCI. The results of the mean of 5-fold cross-validation are shown in Table 2. Our *GNN-PDE-COV* model not only achieved the highest diagnostic accuracy but also demonstrated a significant improvement (paired *t*-test $p < 0.001$) in disease risk prediction compared to other methods. These results suggest that our approach holds great clinical value for disease early diagnosis.

*Third*, we examine the spreading flows of tau aggregates in CN (cognitively normal) and AD groups. As the inward and outward flows shown in Fig. 5(b), it is evident that there are significantly larger amount of tau spreading between sub-cortical regions and entorhinal cortex in CN (early sign of AD onset) while the volume of subcortical-entorhinal tau spreading is greatly reduced in the late stage of AD. This is consistent with current clinical findings that tau pathology starts from sub-cortical regions and then switches to cortical-cortical propagation as disease progresses [24]. However, our *Tau-FlowNet* offers a fine-granularity brain mapping of region-to-region spreading flows over time, which provides a new window to understand the tau propagation mechanism in AD etiology [13].

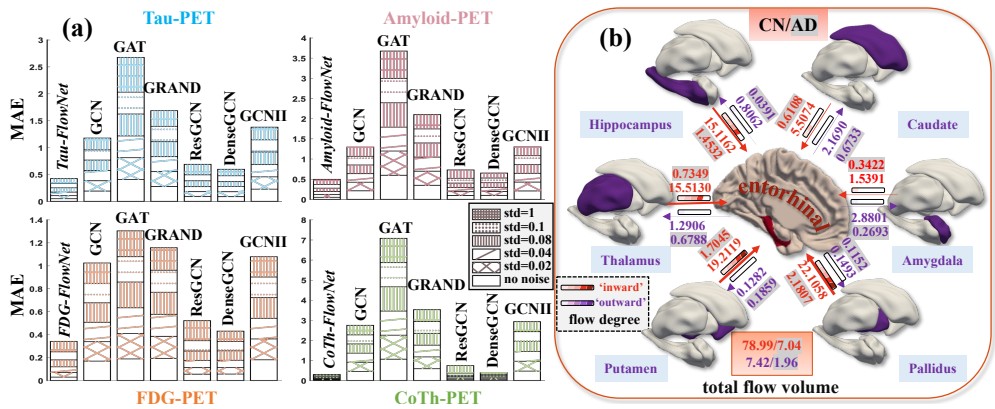

Figure 5: (a) Prediction accuracy by *X-FlowNet* and six benchmark GNN models w.r.t. various noise levels. (b) The subcortical→cortical tau flows are profound in CN. But in AD, there is a diminished extent of such flows.

Table 2: The performance of disease risk prediction. Note: RGCN denotes ResGCN, DGCN denotes DenseGCN. '*' denotes the significant improvement (paired *t*-test: $p < 0.001$). Blue: Tau, red: amyloid, orange: FDG, green: CoTh.

| Tau | Unit (%) | GCN | GCN+ | GAT | GAT+ | GCNII | GCNII+ | RGCN | RGCN+ | DGCN | DGCN+ | GRAND | GRAND+ |
|---|---|---|---|---|---|---|---|---|---|---|---|---|---|
| AD/LMCI | Precision | 80.15 | **90.03(*)** | 69.91 | **86.18(*)** | 83.93 | **90.03(*)** | 84.64 | **89.46(*)** | 84.03 | **91.58(*)** | 87.95 | **88.22(*)** |
| vs. | Accuracy | 82.30 | **88.74(*)** | 81.05 | **87.50(*)** | 83.79 | **88.75(*)** | 86.03 | **90.00(*)** | 85.54 | **91.25(*)** | 88.75 | **90.12(*)** |
| CN/EMCI | F1-score | 75.55 | **84.49(*)** | 72.87 | **84.72(*)** | 78.82 | **84.45(*)** | 83.15 | **88.54(*)** | 82.45 | **91.39(*)** | 88.14 | **89.44(*)** |
| AD | Precision | 89.29 | **91.92(*)** | 87.26 | **90.13(*)** | 83.65 | **88.52(*)** | 92.61 | **95.72(*)** | 92.61 | **95.91(*)** | 91.77 | **95.76(*)** |
| vs. | Accuracy | 86.64 | **90.91(*)** | 84.86 | **88.41(*)** | 76.84 | **86.36(*)** | 91.07 | **95.45(*)** | 91.07 | **95.65(*)** | 90.91 | **95.45(*)** |
| CN | F1-score | 85.64 | **90.26(*)** | 83.99 | **87.16(*)** | 71.51 | **84.68(*)** | 90.45 | **95.32(*)** | 90.45 | **95.55(*)** | 88.86 | **95.38(*)** |
| **Amyloid** | Unit (%) | GCN | GCN+ | GAT | GAT+ | GCNII | GCNII+ | RGCN | RGCN+ | DGCN | DGCN+ | GRAND | GRAND+ |
| AD/LMCI | Precision | 76.36 | **83.78(*)** | 67.73 | **71.79(*)** | 60.87 | **60.01()** | 72.53 | **83.21(*)** | 74.92 | **60.17(*)** | 79.00 | **79.93(*)** |
| vs. | Accuracy | 76.40 | **79.44(*)** | 75.43 | **77.57(*)** | 74.31 | **76.64(*)** | 75.99 | **78.50(*)** | 76.92 | **77.57(*)** | 80.37 | **81.31(*)** |
| CN/EMCI | F1-score | 70.33 | **72.58(*)** | 67.66 | **69.39(*)** | 63.57 | **67.31(*)** | 70.66 | **70.67(*)** | 72.68 | **67.77(*)** | 79.25 | **79.63(*)** |
| AD | Precision | 81.58 | **88.37(*)** | 81.54 | **87.98(*)** | 70.59 | **79.98(*)** | 85.75 | **93.09(*)** | 83.87 | **90.56(*)** | 65.53 | **89.62(*)** |
| vs. | Accuracy | 80.77 | **88.10(*)** | 80.78 | **88.10(*)** | 75.02 | **81.24(*)** | 85.56 | **92.86(*)** | 85.29 | **90.48(*)** | 80.95 | **87.80(*)** |
| CN | F1-score | 78.14 | **87.68(*)** | 78.07 | **87.98(*)** | 65.87 | **77.42(*)** | 85.34 | **92.92(*)** | 82.30 | **90.27(*)** | 72.43 | **88.22(*)** |
| **FDG** | Unit (%) | GCN | GCN+ | GAT | GAT+ | GCNII | GCNII+ | RGCN | RGCN+ | DGCN | DGCN+ | GRAND | GRAND+ |
| AD/LMCI | Precision | 68.43 | **69.29(*)** | 55.86 | **59.29(*)** | 60.08 | **70.94(*)** | 50.45 | **55.14(*)** | 50.45 | **55.14(*)** | 51.38 | **56.25(*)** |
| vs. | Accuracy | 73.17 | **76.00(*)** | 72.17 | **77.00(*)** | 71.78 | **74.54(*)** | 70.98 | **74.26(*)** | 70.98 | **74.26(*)** | 71.10 | **75.00(*)** |
| CN/EMCI | F1-score | 63.94 | **68.15(*)** | 62.15 | **66.99(*)** | 61.02 | **69.07(*)** | 58.96 | **63.29(*)** | 58.96 | **63.29(*)** | 59.42 | **64.29(*)** |
| AD | Precision | 81.11 | **87.25(*)** | 61.90 | **62.33(*)** | 74.31 | **81.06(*)** | 59.77 | **80.57(*)** | 66.84 | **81.77(*)** | 70.91 | **72.24(*)** |
| vs. | Accuracy | 82.17 | **84.62(*)** | 72.82 | **78.95(*)** | 79.55 | **82.05(*)** | 73.35 | **79.58(*)** | 73.87 | **80.11(*)** | 84.21 | **86.32(*)** |
| CN | F1-score | 79.40 | **82.04(*)** | 64.23 | **69.66(*)** | 73.88 | **80.99(*)** | 62.77 | **75.98(*)** | 63.92 | **76.58(*)** | 76.99 | **78.06(*)** |
| **CoTh** | Unit (%) | GCN | GCN+ | GAT | GAT+ | GCNII | GCNII+ | RGCN | RGCN+ | DGCN | DGCN+ | GRAND | GRAND+ |
| AD/LMCI | Precision | 74.85 | **76.23(*)** | 62.63 | **67.15(*)** | 62.63 | **74.71(*)** | 62.63 | **68.77(*)** | 62.63 | **64.59(*)** | 63.81 | **68.77(*)** |
| vs. | Accuracy | 80.68 | **82.32(*)** | 79.10 | **79.34(*)** | 79.10 | **80.37(*)** | 79.10 | **82.93(*)** | 79.10 | **80.37(*)** | 79.88 | **82.93(*)** |
| CN/EMCI | F1-score | 73.55 | **75.93(*)** | 69.89 | **70.44(*)** | 69.89 | **72.72(*)** | 69.89 | **75.19(*)** | 69.89 | **71.62(*)** | 70.94 | **75.19(*)** |
| AD | Precision | 83.45 | **85.77(*)** | 71.24 | **72.80(*)** | 76.14 | **79.84(*)** | 65.04 | **80.62(*)** | 65.04 | **81.52(*)** | 71.24 | **74.37(*)** |
| vs. | Accuracy | 84.79 | **87.16(*)** | 81.50 | **85.32(*)** | 81.50 | **83.49(*)** | 80.59 | **83.52(*)** | 80.59 | **82.42(*)** | 84.40 | **86.24(*)** |
| CN | F1-score | 82.02 | **83.69(*)** | 75.06 | **78.56(*)** | 74.17 | **81.07(*)** | 71.95 | **78.48(*)** | 71.95 | **76.13(*)** | 77.27 | **79.87(*)** |

## 4 Conclusion

In this work, we present the *GNN-PDE-COV* framework to re-think and re-design GNN models with great mathematical insight. On top of this, we devise the selective inductive bias to address the over-smoothing problem in GNN and develop new GNN model to predict the pathology flows *in-vivo* via longitudinal neuroimages. Future work may involve exploring innovative graph regularization techniques and conducting further validation on a broader range of graph-based learning tasks.

## 5 Acknowledgement

This work was supported by the National Institutes of Health AG070701, AG073927, AG068399, and Foundation of Hope. Data collection and sharing for this project was funded by the Alzheimer's Disease Neuroimaging Initiative (ADNI) (National Institutes of Health Grant U01 AG024904) and DOD ADNI (Department of Defense award number W81XWH-12-2-0012).

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
