# Re-Think and Re-Design Graph Neural Networks in Spaces of Continuous Graph Diffusion Functionals

**Tingting Dan**[1], **Jiaqi Ding**[2], **Ziquan Wei**[1], **Shahar Z Kovalsky**[3], **Minjeong Kim**[4]
**Won Hwa Kim**[5], **Guorong Wu**[1,2,*]
[1] Department of Psychiatry, University of North Carolina at Chapel Hill
[2] Department of Computer Science, University of North Carolina at Chapel Hill
[3] Department of Mathematics, University of North Carolina at Chapel Hill
[4] Department of Computer Science, University of North Carolina at Greensboro
[5] Computer Science and Engineering / Graduate School of AI, POSTECH
[*] grwu@med.unc.edu

## S1 Solving variational problems: From objective functional to E-L equations

### S1.1 Step-by-step derivation of min-max optimization in Section 2.2.1

By substituting Eq. 2 into Eq. 1 in the main manuscript, we can obtain the objective function of subscript $z$ (we temporarily drop $i$ for clarity):

$$\mathcal{J}(z) = \max_{|z| \leq \mathbf{1}} \left\| \frac{\lambda}{2} z \nabla_{\mathcal{G}} x \right\|_2^2 + \lambda z \nabla_{\mathcal{G}} (x^0 - \frac{\lambda}{2} z \nabla \mathcal{G} x) \tag{S1}$$

$$= \max_{|z| \leq \mathbf{1}} -\frac{\lambda^2}{4} z \nabla_{\mathcal{G}} x z \nabla_{\mathcal{G}} x + \lambda z \nabla_{\mathcal{G}} x^0 \tag{S2}$$

Next, we convert Eq. S2 into a minimization problem as follows:

$$z = \underset{|z| \leq \mathbf{1}}{\arg \min} \quad z \nabla_{\mathcal{G}} x z \nabla_{\mathcal{G}} x - \frac{4}{\lambda} z \nabla_{\mathcal{G}} x^0 \tag{S3}$$

By letting the derivative with respect to $z_i$ to zero, we have the following equation

$$\nabla_{\mathcal{G}} x z \nabla_{\mathcal{G}} x = \frac{4}{\lambda} \nabla_{\mathcal{G}} x^0 \tag{S4}$$

Since $z$ might be in high dimensional space, solving such a large system of linear equations under the constraint $|z| \leq 1$ is oftentimes computationally challenging. In order to find a practical solution for $z$ that satisfies the constrained minimization problem in Eq. S3, we resort to the majorization-minimization (MM) method [3]. First, we define:

$$M(z) = z \nabla_{\mathcal{G}} x z \nabla_{\mathcal{G}} x - \frac{4}{\lambda} z \nabla_{\mathcal{G}} x^0 \tag{S5}$$

By setting $z^l$ as point of coincidence, we can find a separable majorizer of $M(z)$ by adding the non-negative function

$$(z - z^l)^{\mathsf{T}} (\beta I - \nabla_{\mathcal{G}} x \nabla_{\mathcal{G}} x^{\mathsf{T}})(z - z^l) \tag{S6}$$

37th Conference on Neural Information Processing Systems (NeurIPS 2023).

to $M(z)$, where $\beta$ is greater than or equal to the maximum eigenvalue of $\nabla_{\mathcal{G}}x\nabla_{\mathcal{G}}x^\mathsf{T}$. Note, to unify the format, we use the matrix transpose property in Eq. S6. Therefore, a majorizer of $M(z)$ is given by:

$$M(z) + (z - z^l)^\mathsf{T}(\beta I - \nabla_{\mathcal{G}}x\nabla_{\mathcal{G}}x^\mathsf{T})(z - z^l) \tag{S7}$$

And, using the MM approach, we can obtain the update equation for $z$ as follows:

$$
\begin{aligned}
z^{l+1} &= \operatorname*{arg\,min}_{|z|\leq 1}(M(z) + (z - z^l)^\mathsf{T}(\beta I - \nabla_{\mathcal{G}}x\nabla\mathcal{G}x^\mathsf{T})(z - z^l)) \\
&= \operatorname*{arg\,min}_{|z|\leq 1}(\beta z^\mathsf{T}z - 2(\nabla_{\mathcal{G}}(\frac{2}{\lambda}x^0 - \nabla\mathcal{G}xz^l) + \beta z^l)^\mathsf{T}z) \\
&= \operatorname*{arg\,min}_{|z|\leq 1}(z^\mathsf{T}z - 2(\frac{1}{\beta}\nabla_{\mathcal{G}}(\frac{2}{\lambda}x^0 - \nabla\mathcal{G}xz^l) + z^l)^\mathsf{T}z) \\
&= \operatorname*{arg\,min}_{|z|\leq 1}(z^\mathsf{T}z - 2b^\mathsf{T}z)
\end{aligned}
\tag{S8}
$$

where $b = z^l + \frac{1}{\beta}\nabla_{\mathcal{G}}(\frac{2}{\lambda}x^0 - \nabla_{\mathcal{G}}xz^l)$.

Then, the next step is to find $z \in \mathcal{R}^N$ that minimizes $z^\mathsf{T}z - 2bz$ subject to the constraint $|z| \leq 1$. Let's first consider the simplest case where $z$ is a scalar:

$$\operatorname*{arg\,min}_{|z|\leq 1} \quad z^2 - 2bz \tag{S9}$$

The minimum of $z^2 - 2bz$ is at $z = b$. If $b \leq 1$, then the solution is $z = b$. If $|b| \geq 1$, then the solution is $z = sign(b)$. We can define the gathering function $g(\cdot)$ as:

$$g(b, 1) := \left\{ \begin{array}{cc} b & |b| \leq 1 \\ sign(b) & |b| \geq 1 \end{array} \right. \tag{S10}$$

as illustrated in the middle of Fig. 3 in the main text, then we can write the solution to Eq. S9 as $z = g(b, 1)$.

Note that the vector case Eq. S8 is separable - the elements of $z$ are uncoupled so the constrained minimization can be performed element-wise. Therefore, an update equation for $z$ is given by:

$$z^{l+1} = g(z^l + \frac{1}{\beta}\nabla_{\mathcal{G}}(\frac{2}{\lambda}x^0 - \nabla_{\mathcal{G}}xz^l), 1) \tag{S11}$$

where $l$ denotes the index of the network layer, the representation of $(l+1)^{th}$ is given by Eq. (1) in the main manuscript. Because the optimization problem is convex, the iteration will converge from any initialization. We may choose, say $z^0 = 0$. We call this the iterative *diffusion-gathering (DG) algorithm*.

This algorithm can also be written as

$$
\begin{aligned}
x^{l+1} &= x^0 - \frac{\lambda}{2}\nabla_{\mathcal{G}}^\mathsf{T}z^l \\
z^{l+1} &= g\left(z^l + \frac{2}{\beta\lambda}\nabla_{\mathcal{G}}x^{l+1}, 1\right).
\end{aligned}
\tag{S12}
$$

By scaling $z$ with a factor of $\lambda/2$, we have the following equivalent formulations:

$$
\begin{aligned}
x^{l+1} &= x^0 - \nabla_{\mathcal{G}}^\mathsf{T}z^l \\
z^{l+1} &= g\left(z^{(i)} + \frac{1}{\beta}\nabla_{\mathcal{G}}x^{l+1}, \frac{\lambda}{2}\right)
\end{aligned}
\tag{S13}
$$

We summarize the process of the diffusion-gathering (*DG*) layer in Algorithm 1 (it is similar to the iterative shrinkage threshold algorithm [1]):

**Algorithm 1** *DG* layer process

The objective function:

$$\mathcal{J}(x) = \min_x (\|x - x^{(0)}\|_2^2 + \mathcal{J}_{TV}(x))$$

can be minimized by alternating the following two steps:

$$x^l = x^0 - \nabla_{\mathcal{G}} x^{\mathsf{T}} z^{l-1}$$

$$z^l = g\left(z^{l-1} + \tfrac{1}{\beta}\nabla_{\mathcal{G}} x^l, \tfrac{\lambda}{2}\right) = g\left(z^{l-1} + \tfrac{2}{\beta\lambda}\nabla_{\mathcal{G}} x^l, 1\right)$$

for $l \geq 1$ with $z^0 = 0$ and $\beta \geq maxeig(\nabla_{\mathcal{G}} x^{\mathsf{T}} \nabla_{\mathcal{G}} x)$

## S1.2 The step-by-step derivation of min-max optimization schema in Section 2.2.2

According to the introduction of Secction 2.2.2 (Eq. 4 and Eq. 5) in the main manuscript, we summarize the following equations,

$$
\begin{cases}
\frac{dx}{dt} + div(q) = 0 \\
u_i = \phi(x_i) \\
q = \alpha \otimes \nabla u \\
\Delta u = div(\nabla u)
\end{cases}
\xrightarrow{\ \text{derive}\ }
\begin{cases}
\frac{dx}{dt} = -div(q) \\
\frac{du}{dt} = -\phi^{-1} div(q) \\
\frac{du}{dt} = -\phi^{-1} div(\alpha \otimes q) \\
\frac{du}{dt} = -\phi^{-1}(\alpha \otimes \Delta u)
\end{cases}
\tag{S14}
$$

Since the PDE in Eq. 5 in the main manuscript is equivalent to the E-L equation of the quadratic functional $\mathcal{J}(u) = \min_u \int_{\mathcal{G}} \alpha \otimes |\nabla_{\mathcal{G}} u|^2 du$ (after taking $\phi$ away), we propose to replace the $\ell_2$-norm integral functional $\mathcal{J}(u)$ with TV-based counterpart

$$\mathcal{J}_{TV}(u) = \min_u \int_{\mathcal{G}} \alpha \otimes |\nabla_{\mathcal{G}} u| du \tag{S15}$$

We then introduce an auxiliary matrix $f$ to lift the undifferentiable barrier, and reformulate the TV-based functional as a dual min-max functional

$$\mathcal{J}_{TV}(u, f) = \min_u \max_f \int_{\mathcal{G}} \alpha \otimes f(\nabla_{\mathcal{G}} u) du \tag{S16}$$

where we maximize $f$ such that $\mathcal{J}_{TV}(u, f)$ is close enough to $\mathcal{J}_{TV}(u)$. Using Gâteaux variations, we assume $u \to u + \varepsilon a$, $f \to f + \varepsilon b$, and the directional derivatives in the directions $a$ and $b$ defined as $\frac{d\mathcal{J}}{d\varepsilon}(u + \varepsilon a)\big|_{\varepsilon \to 0}$ and $\frac{d\mathcal{J}}{d\varepsilon}(f + \varepsilon b)\big|_{\varepsilon \to 0}$. Given a functional $\mathcal{J}_{TV}(u, f)$, its Gâteaux variations is formulated by:

$$
\begin{aligned}
\mathcal{J}_{TV}(u + \varepsilon a, f + \varepsilon b) &= \int \alpha \otimes [(f + \varepsilon b) \cdot (\nabla u + \varepsilon \nabla a)] du \\
\Rightarrow \frac{\partial \mathcal{J}}{\partial \varepsilon}\bigg|_{\varepsilon \to 0} &= \int \alpha \otimes [(f \cdot \nabla a) + (\nabla u b)] \, du \\
\Rightarrow \frac{\partial \mathcal{J}}{\partial \varepsilon}\bigg|_{\varepsilon \to 0} &= \alpha \otimes f \cdot a - \int \alpha \otimes (a \cdot \nabla f) du + \int \alpha \otimes (b \nabla u) du
\end{aligned}
\tag{S17}
$$

Since we assume either $u$ is given at the boundary (Dirichlet boundary condition), the boundary term $\alpha \otimes f \cdot a$ can be dropped. After that, the derivative of $\mathcal{J}_{TV}(u, f)$ becomes:

$$\frac{\partial \mathcal{J}}{\partial \varepsilon}\bigg|_{\varepsilon \to 0} = -\int \alpha \otimes (\nabla f \cdot a + \nabla u \cdot b) \tag{S18}$$

Since the dummy functional $a$ and $b$ are related to $u$ and $f$ respectively, the E-L equation from the Gâteaux variations in Eq. S18 leads to two coupled PDEs:

$$\begin{cases} \max_{f} \frac{df}{dt} = \alpha \otimes \nabla_{\mathcal{G}} u \\ \min_{u} \frac{du}{dt} = \alpha \otimes div(f) \end{cases} \tag{S19}$$

Note, we use the adjoint operator $div(f) = -\nabla f$ to approximate the discretization of $\nabla f$ [5], which allows us to link the minimization of $u$ to the classic graph diffusion process.

## S2 Experimental details

### S2.1 Implementation details

#### S2.1.1 Hyperparameters & training details

Table S1 lists the detailed parameter setting for several GNN-based models, including *X-FlowNet*, *PDENet*, *GCN*, *GAT*, *ResGCN*, *DenseGCN* and *GCNII*.

In the node classification experiments, we set the output dimension to be the number of classes. We adopt the public fixed split [9] to separate these datasets into training, validation, and test sets. We use the accuracy, precision and F1-score of node classification as the evaluation metrics.

For the ADNI dataset prediction experiment, we set the input and output dimensions to be the same as the number of brain nodes cannot be altered. We use 5-fold cross-validation to evaluate the performance of different methods and measure their prediction accuracy using mean absolute error (MAE). We also conduct an ablation study using a two-step approach. First, we train a model (MLP+GNN) shown in the left panel of Fig. 4 (b) in the main manuscript to predict the potential energy filed (PEF) based on the transport equation, then compute the flows using Eq. S19, followed by a GCN-based model to predict the further concentration level of AD-related pathological burdens. Since the deep model in this two-step approach is also formalized from the PDE, we refer to this degraded version as *PDENet*.

In addition, we conduct a prediction of the risk of developing AD using the baseline scan, which can be regarded as a graph classification experiment. This experiment only uses 2 GCN layers with a hidden dimension as $64$ for all methods, while the remaining parameters follow the node classification experiment (Table S1 top).

In this work, all experiments are conducted on a server: Intel(R) Xeon(R) Gold 5220R CPU @ 2.20GHz, NVIDIA RTX A5000. The code is available at https://github.com/Dandy5721/GNN-PDE-COV.

#### S2.1.2 Data pre-processing on ADNI dataset.

In total, 1,291 subjects are selected from ADNI [7] dataset, each having diffusion-weighted imaging (DWI) scans and longitudinal amyloid, FDG, cortical thickness(CoTh) and tau PET scans (2-5 time points). The neuroimage processing consists of the following major steps:

- We segment the T1-weighted image into white matter, gray matter, and cerebral spinal fluid using FSL software [6]. On top of the tissue probability map, we parcellate the cortical surface into 148 cortical regions (frontal lobe, insula lobe, temporal lobe, occipital lobe, parietal lobe, and limbic lobe) and 12 sub-cortical regions (left and right hippocampus, caudate, thalamus, amygdala, globus pallidum, and putamen), using the Destrieux atlas [2] (yellow arrows in Fig. S1). Second, we convert each DWI scan to diffusion tensor images (DTI) [8].

- We apply surface seed-based probabilistic fiber tractography [4] using the DTI data, thus producing a $160 \times 160$ anatomical connectivity matrix (white arrows in Fig. S1). Note, the weight of the anatomical connectivity is defined by the number of fibers linking two brain regions normalized by the total number of fibers in the whole brain ($\Delta$ for graph diffusion in *X-FlowNet*).

- Following the region parcellations, we calculate the regional concentration level (the Cerebellum as the reference) of the amyloid, FDG, CoTh and tau pathologies for each brain region (red arrows in Fig. S1), yielding the input $x \in \mathbb{R}^{160}$ for training *X-FlowNet*, respectively.

Following the clinical outcomes, we partition the subjects into the cognitive normal (CN), early-stage mild cognitive impairment (EMCI), late-stage mild cognitive impairment (LMCI), and AD groups. To facilitate population counts, we regard CN and EMCI as "CN-like" group, while LMCI and AD as "AD-like" groups. Table S2 summarizes the statistics of the two datasets.

Table S1: Parameters setting on Citation network (top) and ADNI data (bottom). $M$ denotes the feature dimension and $C$ denotes the number of classes. For *Cora* dataset, we set $i = 4$ when network layer $L = 2$, $i = 8$ if $L = 4$, $i = 10$ if $L = 8, 16, 32, 64, 128$. For *Citeseer* dataset, we set $i = 4$ when network layer $L = 2$, $i = 8$ if $L = 4$, $i = 11$ if $L = 8, 16, 32, 64, 128$. For Pubmed dataset, we set $i = 4$ when network layer $L = 2$, $i = 8$ if $L = 4, 8, 16, 32, 64, 128$. The hidden dimension of $l^{th}$ is twice that of layer $(l - 1)^{th}$. Take Cora as an example (8 layers), the dimension of the hidden layer is: $1433 \rightarrow 1024 \rightarrow 512 \rightarrow 256 \rightarrow 128 \rightarrow 64 \rightarrow 32 \rightarrow 16 \rightarrow 7$. After exceeding 8 layers, the number of hidden layers is doubled according to the total network layer.

| Algorithm | Optimizer | Learning rate | Weight decay | Hidden layer | Dropout | Epoch |
|---|---|---|---|---|---|---|
| *GCN* | Adam | 0.01 | $5 \times 10^{-4}$ | $M \rightarrow 2^i \rightarrow ... \rightarrow 2^4 \rightarrow C$ | 0.5 | 1500 |
| *GAT* | Adam | 0.001 | $5 \times 10^{-4}$ | head=8, $M \rightarrow 2^i ... \rightarrow C$ | 0.6 | 2000 |
| *RGCN* | Adam | 0.005 | $5 \times 10^{-4}$ | hidden dimension=64 | 0.1 | 2500 |
| *DGCN* | Adam | 0.001 | $5 \times 10^{-4}$ | hidden dimension=64 | 0.1 | 2500 |
| *GRAND* | Adam | 0.01 | $5 \times 10^{-4}$ | hidden dimension=16 | 0.5 | 200 |
| *GCNII* | Adam | 0.005 | $5 \times 10^{-4}$ | hidden dimension=128 | 0.6 | 2000 |
| *GCN* | Adam | 0.001 | $5 \times 10^{-4}$ | hidden dimension=16 | 0.2 | 500 |
| *GAT* | Adam | 0.001 | $5 \times 10^{-4}$ | head=8, hidden dimension=4 | 0.5 | 800 |
| *RGCN* | Adam | 0.001 | $5 \times 10^{-4}$ | hidden dimension=16 | 0.1 | 500 |
| *DGCN* | Adam | 0.01 | $5 \times 10^{-4}$ | hidden dimension=8 | 0.1 | 500 |
| *GCNII* | Adam | 0.005 | $5 \times 10^{-4}$ | hidden dimension=16 | 0.6 | 1500 |
| *GRAND* | Adam | 0.01 | $5 \times 10^{-4}$ | hidden dimension=16 | 0.5 | 500 |
| *X-FlowNet* | Adam | 1e-4/3e-3 | $1 \times 10^{-5}$ | hidden dimension=16 | 0.5 | 500 |
| *PDENet* | Adam | 0.01 | $1 \times 10^{-5}$ | hidden dimension=16 | 0.5 | 500 |

Table S2: Dataset statistics.

| | Node classification (Citation) | | | | Application on flow prediction (ADNI) | |
|---|---|---|---|---|---|---|
| Dataset | Description | | | | Features | # of subjects (CN/AD) |
| | Classes | Nodes | Edges | Features | Amyloid (160) | 304/83 |
| *Cora* | 7 | 2708 | 5429 | 1433 | Tau (160) | 124/37 |
| *Citeseer* | 6 | 3327 | 4732 | 3703 | FDG (160) | 211/63 |
| *Pubmed* | 3 | 19717 | 44338 | 500 | Cortical thickness (160) | 359/110 |

## S2.2 Experiments on node classification

Fig S2 presents the performance of different evaluation criteria (accuracy, precision, and F1-score) across different network layers for node classification by benchmark GNN model (patterned in dash lines) and the counterpart novel GNN model from our *GNN-PDE-COV* framework (patterned by solid lines), where each row is associated with a specific instance of GNN model. It is evident that our proposed *GNN-PDE-COV* consistently outperforms other methods across different layers, with significantly enhanced degrees in accuracy, precision, and F1-score. Moreover, the GNN model yielded from our *GNN-PDE-COV* framework consistently achieves the highest accuracy on all three datasets. Overall, these results demonstrate the state-of-the-art performance by our *GNN-PDE-COV* framework in graph node classification.

The effect of anti-smoothing by gathering operation is shown in Fig. S3. To set up the stage, we put the spotlight on the links that connect two nodes with different categorical labels. In this context,

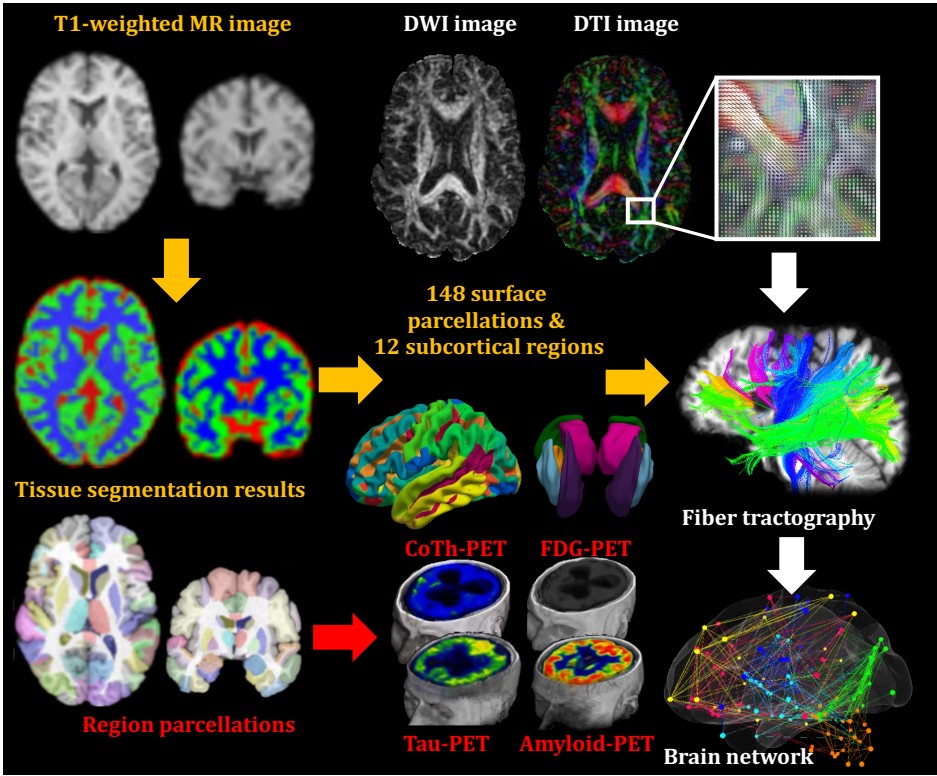

Figure S1: General workflows for processing T1-weighted image (yellow arrows), diffusion-weighted image (white arrows), and PET images (red arrows). The output is shown at the bottom right, including the brain network, and regional concentration level of amyloid, FDG, CoTh and tau aggregates.

2,006 links from *Cora*, 2,408 links from *Citeseer*, and 17,518 links from *Pubmed* datasets are selected, called inter-class links. For each inter-class link, we calculate node-to-node similarity in terms of Pearson's correlation between two associated graph embedding vectors [1] by benchmark methods (in red) and the counterpart GNN models derived from *GNN-PDE-COV* framework (in green). We find that (1) more than 70% nodes are actually associated with inter-class links which confirms the hypothesis of over-smoothing in Fig. 1 of our manuscript; (2) Our novel GNN models have the ability to learn feature representations that better preserve the discriminative power for node classification (as indicated by the distribution of node-to-node similarity shifting towards the sign of anti-correlation).

### S2.3   Application on uncovering the propagation mechanism of pathological events in AD

*Firstly*, we examine the prediction accuracy for each modality of concentration (tau, amyloid, FDG, CoTh) level at different noise levels. Specifically, to evaluate the robustness of our *X-FlowNet* model to noise, we conducted an experiment by adding uncorrelated additive Gaussian noise levels with standard deviation ranging from 0.02 to 1 to the observed concentration levels of tau, amyloid, FDG, and CoTh. We then evaluated the prediction accuracy (MAE) using 5-fold cross-validation. The prediction results, as shown in Fig. S4, indicate that our *X-FlowNet* model is less sensitive to noise added to the imaging features than all other counterpart GNN methods.

*Secondly*, we conduct an ablation study to compare our *X-FlowNet* model with *PDENet* (marked as #7 in Fig. S4). Our model, which is in a GAN architecture and incorporates a TV constraint to avoid over-smoothing, integrates the two steps of estimating the PEF and uncovering the spreading flows into a unified neural network, resulting in significantly improved prediction accuracy compared to *PDENet*.

---

[1]the learned feature representations for node classification

## S2.4 Discussion and limitations

*Discussion.* In our experiments, we found adding *DG* layer right after every FC layer usually does not yield best performance. Instead, we empirically set to add *DG* layer from the first several FC layers. For example, we add *DG* layer after the $3^{rd}$ FC layer in an 8-layer GNN model, after the $5^{th}$ FC layer in a 16-layer GNN model, and after $8^{th}$ FC layer in a GNN model with more than 16 layers. One possible explanation is that the gathering operation in *DG* layer depends on a good estimation of cap $b$ in Eq. 3 (in the main manuscript). Given that the estimation of $b$ may lack stability during the initial stages of graph learning, it can be advantageous to postpone the clip operation from an engineering perspective. However, delaying the addition of the *DG* layer too much can result in missed opportunities to address the problem of over-smoothing.

Regarding the computational time, we record the additional computational time of training our *DG* layer on different datasets. Specifically, the extra training time is 2.2 ms/epoch in *Cora*, 9.8 ms/epoch in *Citeseer*, 7.8 ms/epoch in *Pubmed*, and 0.3 ms/epoch in *ADNI*, respectively, where the data descriptions are listed in Table S2. It is apparent that the TV-based constraint effectively addresses the over-smoothing issue in GNN without imposing a significant computational burden.

*Limitations.* Our current graph learning experiments are limited to citation networks. In the future, we will evaluate our *GNN-PDE-COV* framework on other graph datasets such as drug medicine, protein networks and heterophilic graphs.

*Societal impact.* Our major contribution to the machine learning field is a novel research framework which allows us to develop new GNN models with a system-level understanding. We have provided a new approach to address the common issue of over-smoothing in GNN with a mathematical guarantee. From the application perspective, the new deep model for uncovering the *in-vivo* propagation flows has great potential to establish new underpinning of disease progression and disentangle the heterogeneity of diverse neurodegeneration trajectories.

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

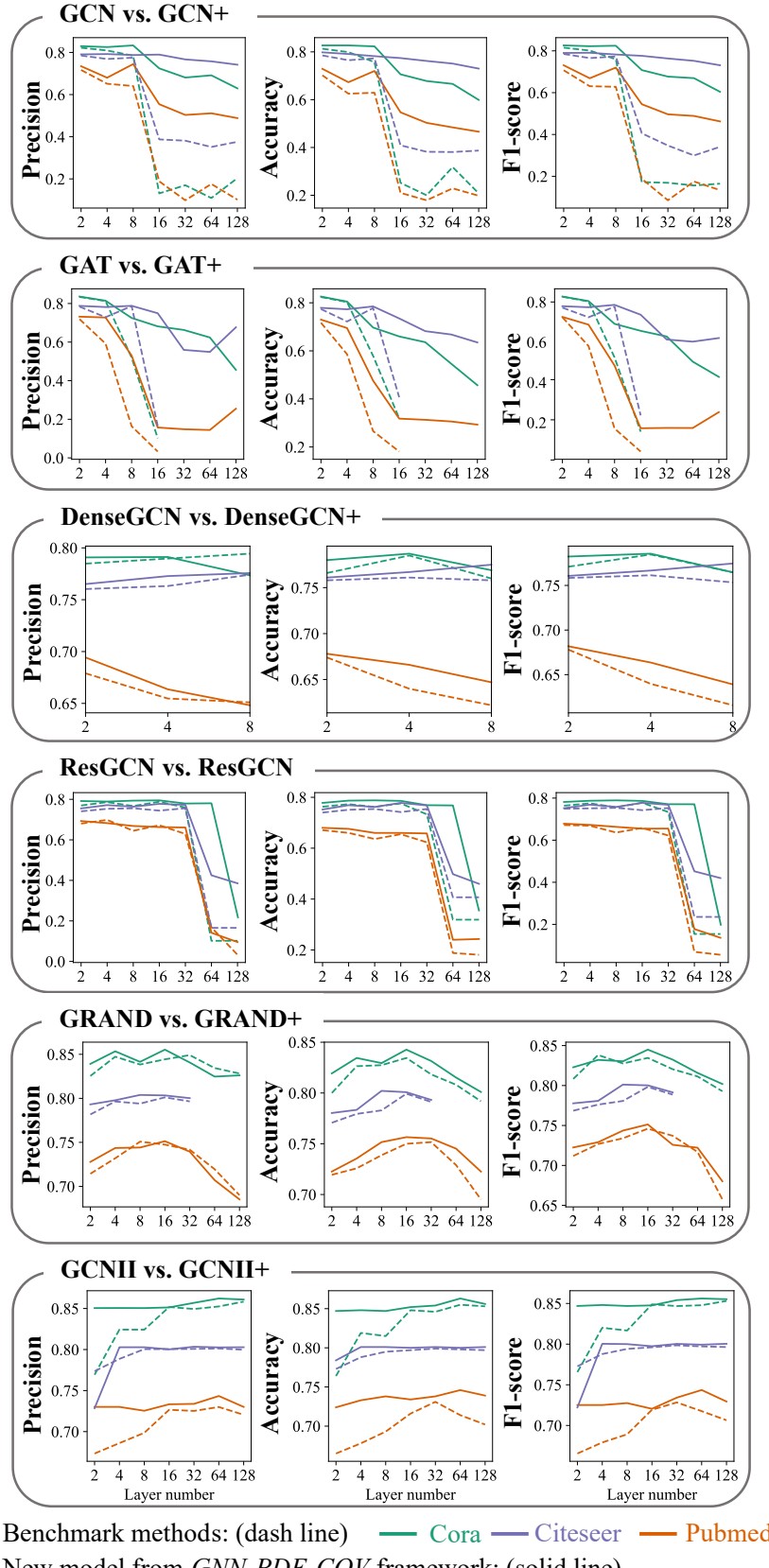

Figure S2: The performance of node classification with respect to various GNN layers (horizontal axis) on six models. Note: dotted line: baseline, solid line: *GNN-PDE-COV*, blue: Cora, purple: Citeseer, red: Pubmed.

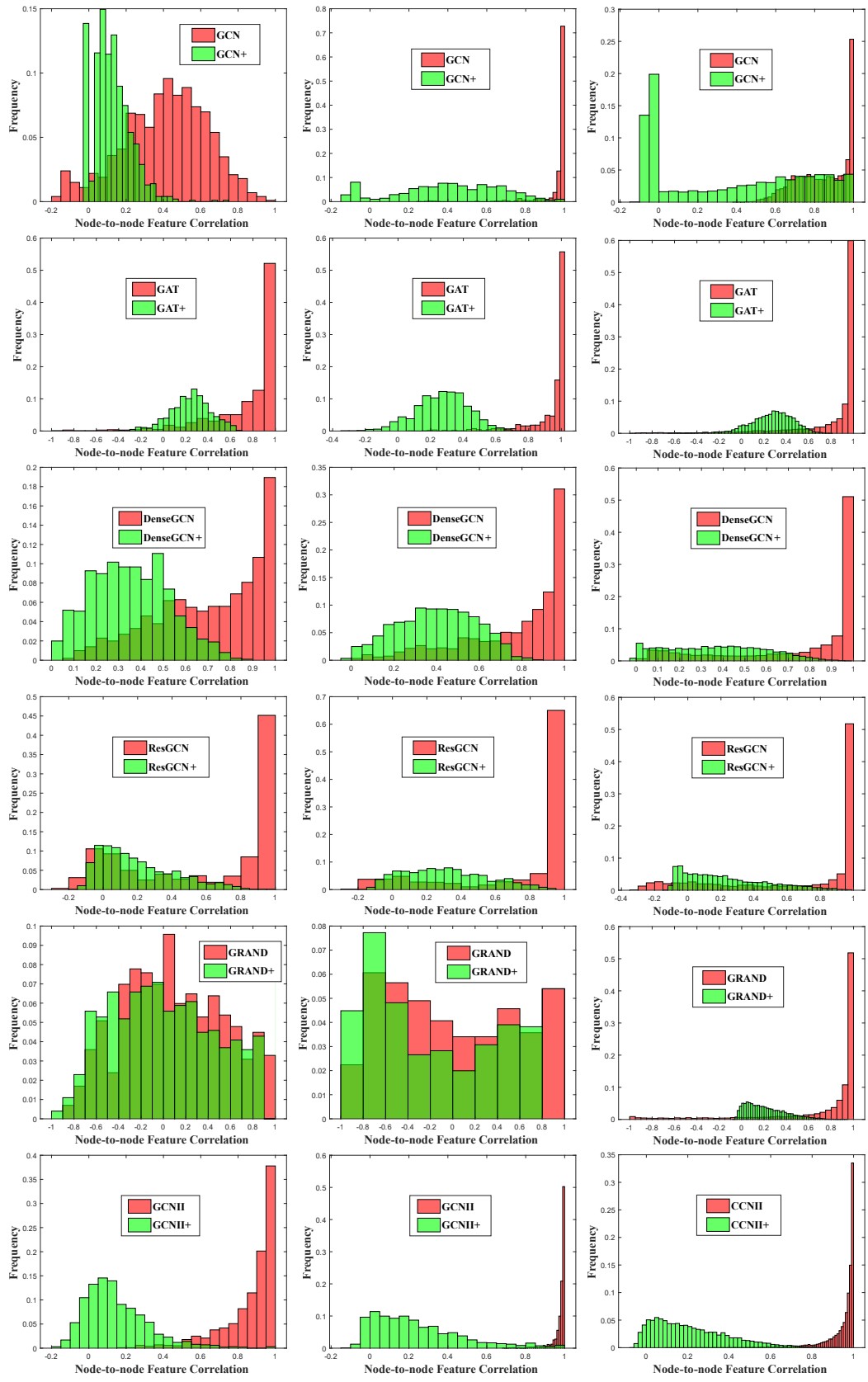

Figure S3: The distribution of the node-to-node similarities (measured by Pearson's correlation between embedding vectors) by Benchmark methods (in red) and our *GNN-PDE-COV* (in green) in *Cora* (left), *Citeseer* (middle), and *Pubmed* (right).

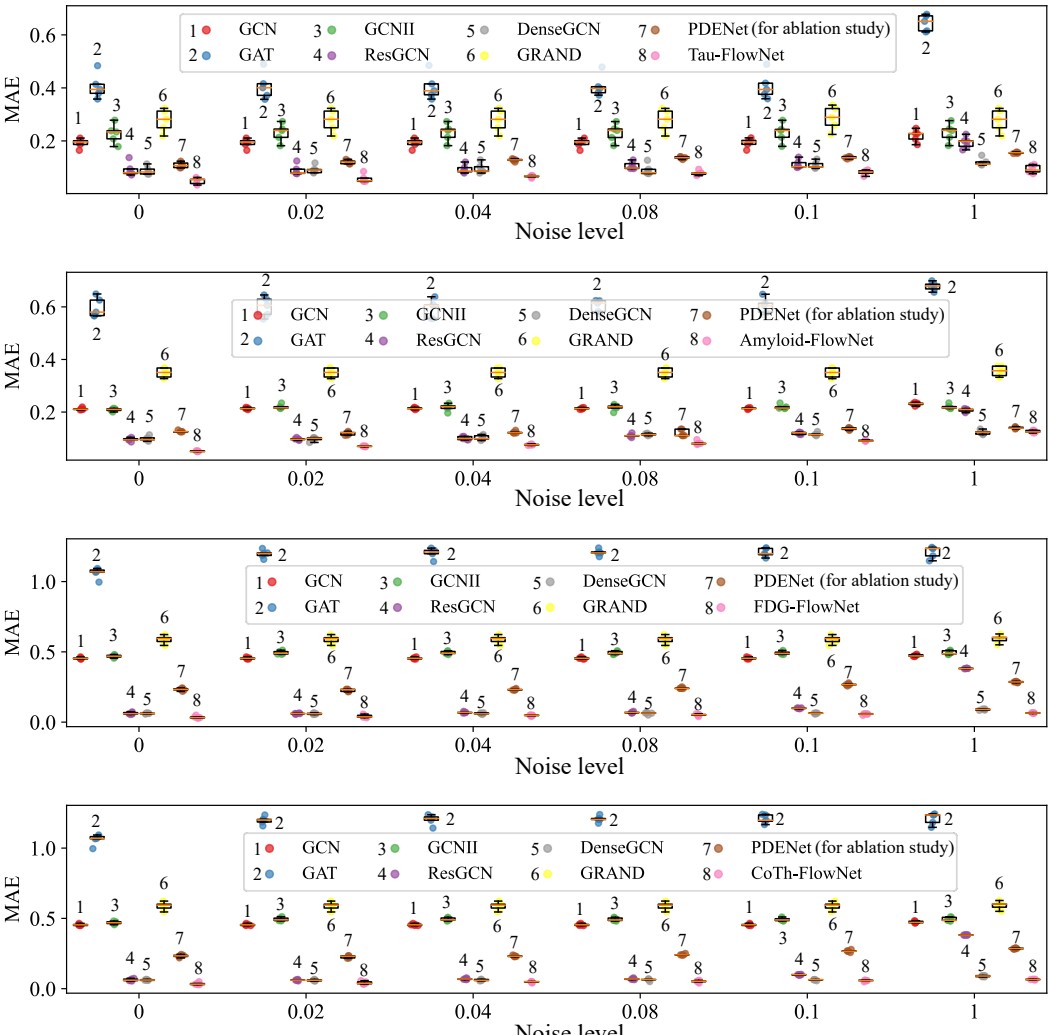

Figure S4: The performance of 5-fold cross-validation for predicting the future concentration (top to bottom: Tau, Amyloid, FDG and CoTh) level by (1) GCN, (2) GAT, (3) GCNII, (4) ResGCN, (5) DenseGCN, (6) GRAND, (7) PDENet (used in ablation study), and (8) our *X-FlowNet*.