# OpenReview forum: "Re-Think and Re-Design Graph Neural Networks in Spaces of Continuous Graph Diffusion Functionals"
_NeurIPS.cc/2023/Conference — NeurIPS 2023 poster_

### Official Review · Reviewer_HWVs · 2023-07-05

**Soundness:** 3 good
**Presentation:** 2 fair
**Contribution:** 2 fair
**Rating:** 4
**Confidence:** 3

**Summary:**

This paper focuses on devising a new inductive bias for cutting-edge graph application and present a general framework through the lens of variational analysis. To this end, the authors first introduce a new selective mechanism that can be easily integrated into existing GNNs and effectively address the trade-off between model depth and over-smoothing, and then devise a novel generative adversarial network to predict the spreading flows in the graph through a neural transport equation. Extensive experiments show that the proposed GNN models can achieve state-of-the-art performance on graph learning benchmarks such as Cora, Citeseer, and Pubmed.

**Strengths:**

1. The perspective of the article is innovative, as it creatively utilizes heuristic problems from traditional physics to inspire the design of GNN structures.
2. The description of the proposed GNN architecture is clear, and the experimental setup is well-defined.
3. The experimental results validate the superiority of the method proposed by the author and demonstrate the rationality of the proposed theory.


**Weaknesses:**

1. The majority of the content in the article is based on the Euler-Lagrange (E-L) equation of the heat kernel. However, I don't believe that the detailed knowledge of the EL equation is familiar to every graph neural network researcher. Therefore, I think it is important to briefly introduce it when it first appears (Line 74) or mention it in the appendix. This is crucial for maintaining the readability and coherence of the article. In fact, it took me a lot of time to consult relevant materials for the subsequent introduction of the Euler-Lagrange (E-L) equation to LaGrangian Mechanics.
2. The issue of over-smoothing in GNNs has been extensively studied since 2020, but the theoretical reasons behind this problem have not been conclusively determined [1, 2, 3]. In line 135-136, the authors claim that after connecting the GNN inductive bias to the function of the graph diffusion process, we can postulate that the root cause of over-smoothing is the isotropic regularization mechanism encoded by the ℓ2-norm. Treating this theory as a conclusion is certainly not a problem. However, due to the fact that many subsequent formula derivations are based on this theory, I have to consider whether this theory has been somewhat hasty and overclaimed. I think the authors should provide some deductions before presenting this theory to maintain logical rigor.
3. In line 166, the authors claim that Eq. 1 is the dual formulation with min-max property for the TV distillation problem. It is well known that the dual problem is strictly mathematically defined. I do not fully understand how the duality problem here is derived. Please provide further explanation.
4. In line 185-186, I am confused about how the recursive min-max solution for Eq. 1 is obtained by disentangling the building block in vanilla GNN into the feature representation learning and graph diffusion underling TV. I do not intuitively perceive the connection between the two. Please explain.

reference:
1. DropEdge: Towards Deep Graph Convolutional Networks on Node Classification. ICLR 2020
2. Towards Deeper Graph Neural Networks. KDD 2020
3. Beyond Low-frequency Information in Graph Convolutional Networks. AAAI 2021


**Questions:**

Please see the weak points.

**Limitations:**

Please see the weak points.

---

> ### Author Rebuttal · Authors · 2023-08-09
>
> We appreciate this reviewer’s insightful comments. We provide answers to the four major questions below.
>
> 1. $\bf Q$: Lack of detailed knowledge of the EL equation for GNN researchers.
>
>     $\bf A$: We apologize for this oversight. We will add more details and relevant work in the Supplementary. Specifically, we plan to open the background introduction by discussing the NeuroODE work [1], which offers a PDE-based implementation for ResNet. Additionally, we will provide a brief summary of the pioneering contributions [2-4] that establish connections between GNN and heat equations on graph data. Our work draws significant inspiration from these early studies.
>
> [1] Neural ordinary differential equations. NeurIPS 2018.
>
> [2] Grand: Graph neural diffusion. ICML 2021.
>
> [3] GRAND++: Graph neural diffusion with a source term. ICLR 2021.
>
> [4] PDE-GCN: Novel architectures for graph neural networks motivated by partial differential equations. NeurIPS 2021.
>
> 2. $\bf Q$: Should provide some deduction regarding the cause of over-smoothing to maintain logical rigor.
>
>     $\bf A$: We appreciate this constructive comment. We will make it clear in the final version by the following means.
>
> $\bf First$, we will introduce the GNN models in [1-3] (shown below) in the final version (placed in the last paragraph of section 2.1), as part of relevant works. Specifically, [1] showed an interesting engineering solution to alleviate the over-smoothing issue by trimming graph nodes/edges. [2] proposed to disentangle feature learning and propagation steps which is similar to our idea of adding the FC layer and DC layer (in Fig. 3 of the main manuscript). The approach in [3] presented an adaptive information aggregation approach by treating low and high-frequency information differently.
>
> $\bf Second$, we will follow the approach in [2] to explain the intuition of why deeper GNN fails using $l_2$-norm regularization term from the perspective of (1) t-SNE visualization of node feature representations as the number of GNN layers increases, (2) the evolution curve of $l_2$-norm, and (3) the evolution curve of the TV term, as the number of layers increases. Preliminary results on the Cora dataset are shown in Fig. 2-3 of the 1-page PDF. We will show the same results for other datasets (such as PubMed and Citeseer) in the Supplementary of the final version. It is clear that (1) the topological community structure by isotropic diffusion is much less consistent with the label distributions than TV-based adaptive diffusion (in Fig. 2), and (2) the trajectory of $l_2$-norm drops much faster than the counterpart curve by TV term (in Fig. 3), indicating that the effectiveness of TV-based GNN in alleviating the vanishing of graph gradients (effect of the over-smoothing issue).
>
> $\bf Third$, we will strengthen the rigor by linking the classic TV-based work (such as Merriman-Bence-Osher (MBO) scheme [4]) in image processing with our TV-based GNN solution for graph data learning. Despite the distinct mathematical frameworks used to define diffusion processes on grid coordinates and graph structures, they both exhibit a common cause for the over-smoothing issue, which can be attributed to the isotropic regularization mechanism encoded by the $l_2$-norm.
>
> [1] DropEdge: Towards Deep Graph Convolutional Networks on Node Classification. ICLR 2020.
>
> [2] Towards Deeper Graph Neural Networks. KDD 2020.
>
> [3] Beyond Low-frequency Information in Graph Convolutional Networks. AAAI 2021.
>
> [4] An MBO scheme on graphs for classification and image processing. SIAM Journal on Imaging Sciences. 2013;6(4):1903-30.
>
> 3. $\bf Q$: How is the duality problem in Eq. 1 (line 166) derived?
>
>     $\bf A$: We apologize for the confusion. Since the TV term is not differentiable at 0, there are two common approaches to circumvent this issue: (1) replace |.| by a function which behaves almost like the magnitude function while is differentiable, e.g., $|x|=(x^2)/\sqrt{(x^2+\epsilon^2)}$ where $\epsilon$ is a small perturbation; (2) introduce a dual variable and rewrite the minimization problem into the min-max scheme (line 158-159) by introducing the dual variable $z$. We choose to employ the dual formulation (specifically the Lagrange dual form), as commonly used in TV-based image processing [1]. We will make it clear in the final version.
>
> [1] A simple primal–dual method for total variation image restoration. Journal of Visual Communication and Image Representation, 2016, 38, 814-823.
>
> 4. $\bf Q$: How the recursive min-max solution for Eq. 1 is obtained by disentangling the building block in vanilla GNN into the feature
>          representation learning and graph diffusion underlying TV?
>
>
>  $\bf A$: In general, the min-max optimization consists of two alternating steps: (1) solving for $x$ (while $z$ is fixed) by minimizing the $l_2$-norm graph smoothness term (Eq. 2 in line 172) and (2) solving for $z$ (while $x$ is fixed) where we employ the majorization-minimization (MM) method (line 178-181). To solve for $x$, Following early work (such as GRAND) on linking PDE and GNN, Eq. 2 essentially describes a heat kernel diffusion process. Thus, the solution for $x$ can be achieved by the discrete GNN model. Meanwhile, the solution for $z$ boils down to a node-wise clip operation (Eq. 3 in line 180). Since the optimization of $x$ and $z$ have been decoupled into two alternating steps, we encapsulate them into a building block consisting of FC-layer (for solving $x$) and DC-layer (for solving $z$), and then cascade a collection of building blocks into a deep GNN model. We will make it clear in the final version.

---

> ### Comment · Area_Chair_N32h · 2023-08-17
> **Please provide additional feedback**
>
> Hi,
>
> You seem to have a low score for this paper. Could you please acknowledge that you have read the rebuttal and let us know if you still have concerns or not? If not, then I would encourage you to increase your score.

---

> > ### Comment · Reviewer_HWVs · 2023-08-17
> >
> > Thank the author for the detailed rebuttals about Q1, Q2 and Q4.
> >
> > The authors have added more details and relevant work in the Supplementary and provided a brief summary of the pioneering contributions.
> >
> > I notice that Eq1 is important for the coherence of the article. However, for Q3, what I am curious about is the derivation process of the duality problem in Eq. 1 (line 166), since the authors claim that Eq. 1 is the dual formulation with min-max property for the TV distillation problem. I would like to emphasize that the dual problem is strictly mathematically defined clearly. Hence the derivation process is necessary. The author’s response is just the motivation.
> >
> > So I would keep my score.
> >
> > (Sorry for putting my previous reply in the wrong position.)

---

> > > ### Author Response · Authors · 2023-08-19
> > > **Response to Reviewer HWVs**
> > >
> > > Dear, Reviewer HWVs,
> > >
> > > We are glad that we have addressed the concerns in Q1, Q2, and Q4. We also appreciate reviewer HWVs giving us another chance to further clarify the confusion in Q3.
> > >
> > > The derivation process of the duality problem in Eq. (1) is as follows:
> > >
> > > The Rudin-Osher-Fatemi (ROF) model solves the minimization problem:
> > >
> > >  $\mathop {\min }\limits_x  \left\| {x - {x^0}} \right\|_2^2 + \lambda \int |{{\nabla _{\cal \mathcal{G}}}x}| dx$.
> > >
> > > To derive the dual formulation, we recall that the TV-norm can be reformulated as:
> > >
> > > $\mathop \int|\nabla_\mathcal{G} x| dx = \mathop {\max }\limits_{|z| \le 1} \mathop \int \nabla_\mathcal{G} x \cdot z$.
> > >
> > > (please see for instance, Zhu, M., Wright, S. J., & Chan, T. F. (2010). Duality-based algorithms for total-variation-regularized image restoration. Computational Optimization and Applications).
> > >
> > > With this definition, the ROF model becomes:
> > >
> > > $\mathop {\min }\limits_x \mathop {\max }\limits_z \left\| {x - {x^0}} \right\|_2^2 + \lambda \int {{\nabla _{\cal \mathcal{G}}}x} \cdot z dx$,
> > >
> > > where $x$ and $z$ are primal and dual variables, respectively.
> > >
> > > The min-max theorem (Chapter VI, Proposition 2.4 in Ekeland, I., Témam, R.: Convex Analysis and Variational Problems. SIAM Classics in Applied Mathematics. SIAM, Philadelphia, 1999) allows us to interchange the min and max, to obtain
> > >
> > > $ \mathop {\max }\limits_z \mathop {\min }\limits_x \left\| {x - {x^0}} \right\|_2^2 + \lambda \int {{\nabla _{\cal \mathcal{G}}}x} \cdot z dx$.
> > >
> > > Therefore, Eq. (1) is proved.
> > >
> > > The derivation of dual optimization has been widely studied in the literature [1-4]. Since it is not our contribution, we will include the step-by-step derivation in the Supplementary of the final version.
> > >
> > > [1] Carter, J.L.: Dual method for total variation-based image restoration. Report 02-13, UCLA CAM (2002)
> > >
> > > [2] Chambolle, A.: An algorithm for total variation minimization and applications. J. Math. Imaging Vis. 20, 89–97 (2004)
> > >
> > > [3] Chan, T.F., Golub, G.H., Mulet, P.: A nonlinear primal-dual method for total variation based image restoration. SIAM J. Sci. Comput. 20, 1964–1977 (1999)
> > >
> > > [4] Zhu, M., Wright, S.J. Chan, T.F.: Duality-based algorithm for total-variation-regularized image restoration, Computational Optimization and Applications, 47, 377-400 (2010)
> > >
> > > Hope we have addressed your question, thank you so much.
> > >
> > > Best,
> > >
> > > Authors

---

### Official Review · Reviewer_sJUp · 2023-07-05

**Soundness:** 3 good
**Presentation:** 3 good
**Contribution:** 4 excellent
**Rating:** 7
**Confidence:** 3

**Summary:**

This work studies a novel design of GNN leveraging the connection with graph diffusion. Specifically, this work considers the discrete GNN model in view of continuous graph diffusion functional formulated as the Euler-Lagrange equation. This work proposes a new design of GNN with selective inductive bias which alleviates the over-smoothing in GNNs. Further, this work proposes a new approach for predicting the flow dynamics in the graph via a neural transport equation using the GAN model to predict the spreading flow.

**Strengths:**

- The paper is well-written with clear motivation and justification. The writing is easy to follow with summarized questions and the paper's approach (re-think and re-design) helps to understand this work.

- The proposed architecture is novel yet simple based on theoretical justifications leveraging the connection with the graph diffusion (although this was studied widely in other papers). Further, the GAN model for flow prediction is also new, using neural transport equations.

- The experimental results show superior performance compared to the baselines in node classification, and the results in node classification demonstrate that the proposed architecture mitigates the over-smoothing issue.

**Weaknesses:**

- Related work section could greatly help the readers for understanding this work as it deals with new GNN as well as flow prediction in graphs. I assume that the related work section was omitted due to the page limit, and recommend adding it if possible.

- The reason for the superior performance of the GAN model is not clear.

**Questions:**

- What is the purpose of using total variation? Is it simply from the inspiration explained in line 151?

- Although the extra training time is described in section S2.4, comparing it with the training time without DC layer would show that the computational burden is not significant.

**Limitations:**

The limitations are discussed in the supplementary file section S2.4.

---

> ### Author Rebuttal · Authors · 2023-08-09
>
> We appreciate the valuable comments from this reviewer. We will incorporate all suggestions into the final version.
>
> We will add a “Related Work” section in the Supplementary which includes the relevant GNN work in solving over-smoothing issues (suggested by Reviewer HWVs) and neuroimaging background on predicting pathology spreading flows.
>
> In general, the performance gain of GAN is largely from the min-max optimization schema. Specifically, mounting neuroscience evidence shows that the propagation of disease-specific pathologies underlines the community structure of the brain’s wiring diagrams [1, 2]. Following this assumption, we formulate the flow estimation problem into a TV-constrained objective function, where we minimize the graph smoothness constraint (using the graph heat kernels) while maximizing the inter-community flux ($\alpha$ in Eq. 6). After that, we devise the equivalent discrete deep model using GAN. At a higher level, the promising result of flow estimation showcases the effectiveness of our GNN-PDE-COV framework in designing a less “black-box” GNN model for real-world machine learning problem.
>
> [1] Deborah N Schoonhoven and others, Tau protein spreads through functionally connected neurons in Alzheimer’s disease: a combined MEG/PET study, Brain, 2023.
>
> [2] Steward, A., Biel, D., Luan, Y., Brendel, M., Dewenter, A., Roemer, S.N., Rubinski, A., Dichgans, M., Ewers, M. and Franzmeier, N. (2022), Brain network segregation attenuates tau spreading in Alzheimer’s disease. Alzheimer's Dement., 18: e061626.
>
> Regarding the purpose of using total variation, TV has demonstrated its effectiveness in mitigating the problem of excessive smoothing in image denoising and reconstruction. In this work, we tackle a similar challenge related to over-smoothing in graph data. Upon identifying that the root cause of over-smoothing in existing GNN models is linked to the $l_2$-norm graph smoothness term, we conjecture that, similarly to image processing, TV might be an effective solution for over-smoothing in GNN.
>
> Regarding the extra computational cost due to the min-max schema, we have summarized the comparison of running time with other GNN models (such as GCN, GAT, and GRAND) in Table 3 of the attached 1-page pdf file.

---

> > ### Comment · Reviewer_sJUp · 2023-08-17
> >
> > Thank you for the detailed response.
> > I do not find other major concerns. I believe further adding the intuitions related to GAN as well as the backgrounds (including the citations mentioned in other reviewers' comments) would strengthen the presentation of this work.
> > Thereby I would like to keep my score.

---

> > > ### Author Response · Authors · 2023-08-17
> > > **Response to Reviewer sJUp**
> > >
> > > Thank you for your time and careful response.
> > >
> > > Best regards,
> > >
> > > Authors

---

### Official Review · Reviewer_CRUA · 2023-07-06

**Soundness:** 4 excellent
**Presentation:** 3 good
**Contribution:** 4 excellent
**Rating:** 8
**Confidence:** 3

**Summary:**

The authors connects graph neural networks with discretizations of PDEs and connects their limiting behavior with E-L equations. By changing regularization / smoothing term from second order to first order, the authors designed a new E-L equation and its corresponding GNN. The author then present numerical experiments to show their improvement in performance.

**Strengths:**

- The authors tackles the problem of oversmoothing in GNN, which is an essential problem GNN cannot get as deep as other neural networks.
- The authors provided solid mathematical support on their method and great experimental performance.
- It is an original work connecting TV-minimization from the old school machine learning, with the SOTA design of GNNs.

**Weaknesses:**

- The authors should do an table outlining the steps of their algorithm. It will make it easier for audience who prefer testing model before going through the math.
- The infomation in figure 5 is too dense. It would be better if the authors could split it into two figures.

**Questions:**

- Equation 3 sounds similar to the MBO scheme, which is basically diffusion + clip. Are there any connections in between?
- Does your model transfer to other types of neural networks like convolutional neural networks?
- Does your model requires more computation time? How much more compared to the origional run time of GCN, GAT, and GRAND?

---

> ### Author Rebuttal · Authors · 2023-08-09
>
> We are glad that the reviewer is enthusiastic about this work. We will outline the steps of our algorithm in a table (likely in the Supplementary material). Also, we appreciate the comment regarding the layout of Fig. 5. We will split it into two figures in the final version.
>
> We agree with this reviewer that our min-max optimization is very close to the classic Merriman-Bence-Osher (MBO) scheme for TV-based image filtering. Actually, our work is greatly inspired by these pioneering works.
>
> Our current work only applies to graph data. Recall the connection between ResNet and NeuroODE, it is possible to devise a new CNN-based backbone using the similar framework, that is, designing new functionals in a continuous domain and developing the equivalent discrete deep models.
>
> Compared to GCN and GAT, there is no significant increase of running time by our method since the operations in the DC-layer are element-wise multiplication and clip. However, our method is almost 5 times faster than GRAND which relies on a PDE solver. We have summarized the comparison of running time (by text) in the Supplementary (lines 161-165). We will show the detailed computation time in a table (as shown in Table 3 of the 1-page pdf file in the rebuttal) in the final version.

---

### Official Review · Reviewer_paGB · 2023-07-06

**Soundness:** 3 good
**Presentation:** 1 poor
**Contribution:** 3 good
**Rating:** 6
**Confidence:** 2

**Summary:**

The authors propose a framework based on the Euler-Lagrange equation to derive specialized GNNs by discretizing continuous diffusion functionals. By deriving a new GNN layer from the Total Variation functional, they manage to control the oversmoothing problem in existing GNN architectures and improve node classification performance of six existing architectures with up to 128 layers. Additionally, the authors introduce a new GAN to learn flow problems on graphs and evaluate it on longitudinal data from Alzheimer's disease.

**Strengths:**

The prominent strengths of this paper are the thorough evaluation and the great results on node classification showing that the proposed DC layer can improve different architectures and enable deeper graph neural networks. The proposed GAN also performs very well on flow prediction in an Alzheimer's disease dataset though I cannot judge the subject-specific interpretation of those results. In the derivation of their method, the authors combine many techniques though it is in parts difficult to follow (see Weaknesses).

**Weaknesses:**

The one major weakness of this paper is its (lack of) clarity in writing. Many ideas are insufficiently explained and notation is often inconsistent and confusing. In particular, I refer to the following:

1. Line 90: The numbers for Cora and Citeseer are not in Table 1.
1. Line 110: Graph divergence operator is never defined and no reference is given.
1. Line 111-121: Unclear writing that mixes two different ideas. The first two sentences talk about an often-used (citations missing) regularization term and its effect, but the remainder of the paragraph is about interpreting GNN structures as neural ODEs on graphs.
1. Line 127-130: Drawing the conclusion that you have established this mapping requires either further explanation or at least a citation.
1. Line 136: Which L2 norm does this refer to? The one in the functional in line 127?
1. Line 158-159: The definition of $J_{\mathrm{TV}}$ is completely unclear. $x$ is triple bound (parameter of $J_{\mathrm{TV}}$, parameter of $\min$ operator and integrated over) and $z$ is double bound.
1. Line 160-161: What exactly does this "trick" consist of and in which sense is "degree" used in this sentence?
1. Equation (1): Same issue as line 158-159.
1. Equation (3): Is the clipping elementwise?
1. Line 190: Why are the $x_i$ being clipped when Equation (3) referes to the $z_i$?
1. Line 191: How does large degree increase $z_i$? An arbitrary number of edges of identical nodes would leave $z_i$ still unchanged, wouldn't it?
1. Line 192-193: What does it mean to "shift the diffusion pattern"? Which cases of the case distinction in Equation (3) correspond to "heat-diffusion within community" and "penalizing inter-community exchange"?
1. Line 211: How is future predictability related to the Brachistochrone problem that asks about the shortest path?
1. Section 2.2.2: The whole section is difficult to follow because it introduces yet another problem and solution in little space. While I understand the importance of this section as an example for an alternative GNN architecture derived from the proposed method, for the overall clarity of the paper the space might be better used to clarify the main method.
1. Overall, the importance of the Brachistochrone problem is overemphasized and diverts attention from the main contribution of the paper. It is a nice example application of the E-L equation but even in Figure 1 the analogy between the mechanics and ML case is weak.

This paper leaves the impression of solid work, though the opaque presentation prevents me from saying so with any certainty. A thorough rewrite of Section 2 with a focus on the reader could make this a great paper.

**Questions:**

1. In how far does GNN-PDE-COV rely on the homophily assumption? Part of the problem illustrated in Figure 1 is that the strong connection between nodes of separate classes breaks the homophily assumption of many GNN models.
1. How does the runtime of your model scale in the number of nodes $N$ and number of edges $E$?

**Limitations:**

The appendix mentions briefly limitations of the mode of evaluation. I would be interested in a point on how feasible it is to derive specialized architectures based on the GNN-PDE-COV framework given that the optimization problem in the proposed GAN is completely different from the optimization problem in Section 2.2.1.

---

> ### Author Rebuttal · Authors · 2023-08-09
>
> We appreciate the constructive comments from this reviewer. We answer the general question first and then clarify each specific comment.
>
> General questions:
>
> We appreciate the insightful question regarding the homophily assumption. In our GNN-PDE-COV framework, similar to current GNN models, we do adopt the homophily assumption, which implies that graph embeddings between two nodes that are topologically connected are expected to be similar. As we explained in lines 191-196, we introduce the total variation (TV) on graph gradients to incorporate a high-level heuristic of community structure into the information exchange on the graph. Thus, the homophily assumption remains a fundamental driving factor in our GNN models. The inclusion of the TV term in the DC layer (Eq. 3) provides a flexible mechanism to adaptively control the information exchange based on the global topology of the network community. In the 1-page PDF attached with this rebuttal, we have shown the new benchmark results on five heterophilic graph datasets (Texas, Wisconsin, Actor, Squirrel, Cornell, and Chameleon) in Table 2. The results indicate that our TV-based GNN model also performs well on heterographic data.
>
> Regarding the running time (summarized in Table 3 of the attached 1-page PDF), the driving factor is the number of nodes $N$. This is because the majority of operations in our GNN model, as well as in other GNN methods, are primarily applied to nodes rather than edges.
>
> Specific comments in “Weaknesses” session:
>
> 1. The accuracy numbers for Cora and Citeseer are in Table 1 (128 layers, last column and last row of each dataset). Since the numbers in Table 1 are densely packed, this reviewer might have missed the %. We will highlight them in the final version.
>
> 2. We will explain the graph divergence operator in the final version.
>
> 3. We will smooth out the write-ups to transit from GNN regularizer to PDE. Thanks for this constructive comment.
>
> 4. We will add a reference since it is not our major contribution. Thanks.
>
> 5. Correct. We will make it clear in the final version.
>
> 6. We apologize for this notation issue. We will fix this in the final version by replacing “$J_{TV}(x,z)$” with "$\mathop {\min }\limits_{x} \mathop {\max}\limits_{z}J_{TV}(x,z)$", where $J_{TV}$=...”.
>
> 7. We will include a reference to explain the approximation used to handle the absolute value operator on $z$ during optimization. Due to the page limit, this information will be provided in the Supplementary material.
>
> 8. We have explained in #6.
>
> 9. Correct, the clip operation is element-wise.
>
> 10. We apologize for the confusion. Actually, $z$ is the intermediate result of $x$. Precisely speaking, we update $z$ based on $x$ and then apply clip operation on $z$. After that, we replace $x$ with the updated $z$ for graph diffusion. We will make it clear in the final version.
>
> 11. Following the homogeneity assumption that nodes within the same community share the group-specific characteristics of graph embeddings, inter-community links are supposed to have large gradient gradients, which leads to the large value of $z$.
>
> 12. We have specifically discussed the TV-based diffusion patterns in Supplement S2.2. Please also refer to Fig. S3. Specifically, we find that (1) more than 70% nodes are actually associated with inter-class links which confirm the hypothesis of over-smoothing in Fig. 1 of our manuscript; (2) Our novel GNN models have the ability to learn feature representations that better preserve the discriminative power for node classification (as indicated by the distribution of node-to-node similarity shifting towards the sign of anti-correlation).
>
> 13. We used the shortest path in the Brachistochrone problem as an example motivating work. For each learning problem, the underlying questions may vary. For example, the FlowNet (in 2.2.2) is seeking for a set of max-flows that underline the network community structure.
>
> 14. Due to the page limit, we moved the background of neuroimaging application to the Supplement. We will make it clear in the final version (by adding more detail of background in the Supplementary).
>
> 15. We would like to emphasize that the Brachistochrone problem in Fig. 1 is only used to help readers understand the motivation of linking GNN to a calculus of variation problem. Our major contribution is to address the over-smoothing issue in GNN from the perspective of designing application-specific graph diffusion patterns.
>
> Thank you for your valuable comments, we will incorporate all the comments and suggestions in the final version.

---

> > ### Comment · Reviewer_paGB · 2023-08-10
> >
> > Thank you for the clarifications. Two more point from my side:
> >
> > 1. As the text in line 90 refers to state-of-the-art results, I only checked the red numbers in the table. It should be clarified in the text that you refer to the 128 layer numbers.
> > ---
> > 4. Which paper will you cite here?

---

> > > ### Author Response · Authors · 2023-08-10
> > > **Response to Reviewer paGB**
> > >
> > > Thank you for your careful reply.
> > >
> > > For 1:
> > >
> > > OK, we will make it clear in the final version.
> > >
> > > For 4:
> > >
> > > We apologize for the short answer. In the final version, we will add one sentence in line 130 as:
> > >
> > > “…we have established a mapping between the mechanics of GNN models and the functional of graph diffusion patterns in a continuous domain. Note, similar works can be found in [1,2]. ”
> > >
> > > [1] GRAND: Graph neural diffusion. In: International Conference on Machine Learning (ICML), 2021.
> > >
> > > [2] PDE-GCN: Novel Architectures for Graph Neural Networks Motivated by Partial Differential Equations, Advances in neural information processing systems (NeurIPS), 2021

---

> > > > ### Comment · Reviewer_paGB · 2023-08-13
> > > >
> > > > Based on your response and the clarifications from the other reviews, I raise my score and confidence slightly.

---

> > > > > ### Author Response · Authors · 2023-08-13
> > > > > **Response to Reviewer paGB**
> > > > >
> > > > > Thank you for your time and consideration.
> > > > >
> > > > > Best regards,
> > > > >
> > > > > Authors

---

### Official Review · Reviewer_VF1D · 2023-07-27

**Soundness:** 3 good
**Presentation:** 2 fair
**Contribution:** 2 fair
**Rating:** 3
**Confidence:** 4

**Summary:**

The authors develop a framework for linking discrete (message passing) GNNs to continuous graph diffusion functional networks using Euler-Lagrange equations of heat kernels.  Via this framework, they analyze the causes of oversmoothing in current GNNs. By noting that the main cause of oversmoothing is the minimization of the quadratic graph smoothness term in the diffusion equation, their main contribution is to replace this by a Total Variation (TV) term (as used in image reconstruction, restoration etc.), yielding a new objective function. Further since the $l_1$ norm term of TV is non-differentiable at 0, they develop a dual min-max approach for solving this objective function iteratively by first minimizing for X and then maximizing for Z. This solution technique then results in a new GNN architecture with a separate feature learning layer (Fully Connected layer) for $l^{th}$ layer embeddings $X^l$ followed by a diffusion clip layer for generating the Z terms. They then apply their method for a specific application of predicting spreading flow dynamics. Experimental results are presented on 3 benchmark citation datasets and 6 GNN models including Vanilla GCN and  one other diffusion GCN (GRAND).

TL;DR takeaway of the problem setup/motivation is how to preserve community/contextual embeddings of adjacent (dissimilar) boundary nodes in different labeled communities by preventing oversmoothing through localized message passing. The proposed solution is via the Total Variation parameter Z that should penalize inter community information diffusion.


**Strengths:**

1.	Avoiding local oversmoothing by replacing the quadratic graph smoothness term in the diffusion equation by an $l_1$ norm regularize which is the TV term. This seems to be a novel application of TV to diffusion GNNs.

2.	Formal derivation of an iterative dual min-max method for solving the non-differentiable objective function with the TV term. While this type of iterative technique has gained increasing popularity starting with ADMM, the detailed derivation is a good contribution.

3.	For small layers (2 layers), their method shows performance improvement over the other baselines. This might be due to ameliorating the oversmoothing process due to the TV Z parameter.


**Weaknesses:**

1.	I am surprised the authors do not have references to classic papers that analyze the root causes of oversmoothing in GNNs, for example, the DropEdge paper [1] “Tackling Over-Smoothing for General Graph Convolutional Networks”, W Huang∗ , Yu Rong∗  et al.. IEEE TPAMI Aug. 2015. This paper analyzes oversmoothing by looking at a spectral analysis of the underlying adjacency matrix. It would be interesting to see if there is a connection between the $beta$ parameter and degree connectivity of equation (3) and results from [1].

2.	The baseline model comparisons are too limited - vanilla GCN, GAT, one diffusion GNN (GRAND).
	a. There have been several papers that either explicitly focus on deep sampling while mitigating oversmoothing e.g., [2] “Decoupling the Depth and Scope of Graph Neural Networks”, H. Zeng, M. Zhang et al. Neurips 21.
	b. They test their model on citation networks which are considered highly homophilic. However, dissimilar nodes (as they illustrate in fig. 1) that would ideally prove their oversmoothing claims are adjacent to each other primarily in heterophilic networks. It would be useful if they can show their results on heterophilic graphs.
There needs to be an extended comparison with GNN  models  such as [2] as well as others that look at heterophilic graphs e.g., $H_2GCN$ [3] “Beyond Homophily in Graph Neural Networks: Current Limitations and Effective Designs,” J. Zhu, Y. Yan et al, Arxiv. As they claim their method prevents over-smoothing, it would be very valuable to see comparative results  on heterophilic graphs and if they can show improvements over other models.

3.	Experimental Results:  [3] $H_2GCN$, [4] GeomGCN [5] GPRGNN   seem to show better results. For instance, accuracy of this paper on Pubmed is 80% while H2GCN, GeomGCN, GPRGNN show 90% accuracy. Similar results for Cora and Citeseer .
[4] Geom-GCN: “Geometric Graph Convolutional Networks. In International Conference on Learning Representations”, Hongbin Pei, Bingzhe Wei, Kevin Chen-Chuan Chang, Yu Lei, and Bo Yang. ICLR 2020.
[5] GPRGNN: “Adaptive Universal Generalized PageRank Graph Neural Network”, ICLR2021.

4. There may be some typos in experimental results -  published results are not consistent with published tabular data in the previous works.  E.g. GCNII results for Pubmed are 90% in Table 1 in the following nips 22 paper: https://papers.nips.cc/paper_files/paper/2022/file/75c45fca2aa416ada062b26cc4fb7641-Paper-Conference.pdf but Table 1 in this paper only shows around 79% for GCNII on Pubmed. Similarly please check Table 5 in the following nips 20 paper: https://arxiv.org/pdf/2006.11468.pdf for other discrepancies. I suggest rerunning experiments and checking for typos or otherwise undertanding why numbers are different. In general, the margin of improvement is around 2-4 points over vanilla GCNs for $H_2GCN$ and GPRGNN while the margin is also in that range for this paper, even though they show lower scores for vanilla GNN and GAT. It would definitely improve this paper if you added these papers as baselines for direct comparison.

5. The diffusion flow application that is used in the paper to validate the diffusion model seems to a very niche application. It would greatly add value to the paper if you validated the model over a realistic flow model such as a traffic flow problem.

6. The authors note that their iterative solution technique is not computationally intensive (appendix). However, the method was tried only on 4 limited datasets that are all citation networks. In general, for any iterative solution process the tradeoff between convergence time and accuracy requires more sophisticated and detailed evaluation. Can you improve accuracy on Pubmed (see above) at the cost of increased computation time?








**Questions:**

Please consider different graph setups to validate your oversmoothing diffusion models such as heterophilic graphs in which dissimilar nodes are close by and similar nodes can be far away.

More extensive evaluation of tradeoffs between convergence time and accuracy would be helpful, especially to improve accuracy on the 4 datasets as compared to the newer baselines suggested above.

Consider a more realistic diffusion flow application to validate your diffusion model such as a traffic flow problem.

**Limitations:**

The authors have addressed the limitations of this work in the appendix, primarily the lack of diversity in evaluation datasets. In general, as pointed earlier, paper could be improved with more extensive comparisons on diverse datasets and models.

The authors have stated that the proposed topic doesn't have any negative societal impacts. To the contrary they state that "From the application perspective, the new deep model for uncovering the in-vivo propagation flows has great potential to establish new underpinning of disease progression and disentangle the heterogeneity of diverse neurodegeneration trajectories." While technical correct, this comment in general applies to any work on sensitive datasets in the medical field.

---

> ### Author Rebuttal · Authors · 2023-08-09
>
> We are sincerely thankful for all constructive comments provided by this reviewer. We are pleased that the reviewer recognizes the merits of our new GNN technique, as the major concerns are centered on more extensive comparisons on diverse datasets and models. We have prepared all necessary results in the attached pdf file, which includes comparisons with more GNN models (such as ShaDowGNN [2], $H_2$GCN[3], and GPRGNN [5]) on six new datasets and two new benchmark experiments on traffic flow data. These results are ready to be included in the final version.
>
> $\textbf {(1)}$ Regarding the relatively low accuracy on Cora, Citeseer, and PubMed shown in our paper compared to other published works (weakness #3,4), we would like to emphasize that such discrepancy is due to the use of different data splitting strategies. As shown in Table 1 of the 1-page PDF, we show the classification accuracies by ShaDowGNN, $H_2$GCN, and GPRGNN, utilizing a 3:1:1 (i.e., randomly split nodes of each class into 60%, 20%, and 20% for training, validation and testing, and measure the performance of the models on the test sets over 10 random splits) data split (often used for full-supervised learning) [4] (you mentioned) and a 1:25:50 (with 20 nodes per class for training, 500 nodes for validation and 1000 nodes for testing) split (often used for semi-supervised learning) [11] (as shown in Supplementary file). The relatively higher accuracy referenced by the reviewer pertains to the models trained and tested on the 3:1:1 split, where training data significantly outnumbers the validation and testing sets. In contrast, the results presented in the manuscript are based on the more challenging 1:25:50 split of the dataset.
>
> [11] Revisiting semi-supervised learning with graph embeddings, ICML, 2016
>
> $\textbf {(2)}$ Regarding the additional evaluations on heterophilic graphs and more baseline models (weakness #2,6), we have included Texas, Wisconsin, Actor, Squirrel, Chameleon, and Cornell datasets, where the homophily ratios are less than 0.3. In Table 2 of the 1-page PDF, we have shown the node classification results as the number of GNN layers increases from 16 to 128 (we will add all the experimental results of all the layers (2 to128) on all methods in the final version). It is clear that our TV-based GNN model (GCNII+ by adding DC-layer on top of GCNII backbone) outperforms $H_2$GCN, GPRGNN, ShaDowGNN in all testing scenarios. Note, we observed a consistent 1-3% enhancement over GCNII, which is consistent with the benchmark results we have presented in Table 1 of the manuscript. Also, we are unable to make Geom-GCN running in our GPU environment (with a newer version of deep graph library) since Geom-GCN reportedly requires CUDA 9.2. Given the short turnaround time, we didn’t include Geom-GCN in Table 2 of the attached 1-page PDF. We have released all the codes and data on GitHub (please use the same anonymous link shown in the manuscript if can access).
>
> $\textbf {(3)}$ Regarding the validation of flow experiment on traffic data (weakness #5), we have evaluated our FlowNet on two benchmarks of traffic flow datasets: METR-LA (arXiv:1707.01926) and PEMS-BAY (arXiv:2108.09091). The MAE by our FlowNet is 3.411 in METR-LA (3.229 by the best model) and 1.814 in PEMS-BAY (1.790 by the best model), respectively. Compared with the results published on website paperswithcode (please refer to Fig.1 in the 1-page PDF), our proposed method can achieve competitive promising results, compared to the current state-of-the-art models. Thank you for this constructive suggestion, we will add these experiment results to the Supplementary in the final version.
>
> $\textbf {(4)}$ Regarding the methodology comparison with DropEdge [1] (weakness #1), we appreciate this valuable information. We will include this work in the final version, as part of the related work. Since the journal paper this reviewer is referring to has not yet been published in TPAMI, we will cite their conference paper (DropEdge: Towards Deep Graph Convolutional Networks on Node Classification. ICLR 2020). Although both works aim to address the over-smoothing issue in GNN, the two approaches are completely different. “DropEdge” focused on preventing over-smoothing by reducing the number of network edges which might undermine the graph topology. Our work provided a top-down mathematical framework to regulate the message exchange through a min-max optimization schema. In the perspective of methodology, we would like to argue that there is NO DIRECT connection between the beta parameter in Eq. 3 and the “DropEdge” operation in their work.
>
> $\textbf {(5)}$ Regarding the computational cost (weakness #6), we have summarized the running time for various GNN models in Table 3 of 1-page PDF.
>
> $\textbf {(6)}$ Regarding the concern on societal impact, we mainly use neuroimaging data and clinical outcomes from public databases. Note there are no biological/chemical resources included in this paper. The human subject information of all imaging and demographic data has been completely removed from the public databases.

---

> > ### Comment · Reviewer_HWVs · 2023-08-17
> > **Comments**
> >
> > Thank the author for the detailed rebuttals about Q1, Q2 and Q4.
> >
> > The authors have added more details and relevant work in the Supplementary and provided a brief summary of the pioneering contributions.
> >
> > I notice that Eq1 is important for the coherence of the article. However, for Q3, what I am curious about is the derivation process of the duality problem in Eq. 1 (line 166), since the authors claim that Eq. 1 is the dual formulation with min-max property for the TV distillation problem. I would like to emphasize that the dual problem is strictly mathematically defined clearly. Hence the derivation process is necessary. The author’s response is just the motivation.
> >
> > So I would keep my score.

---

> > ### Comment · Reviewer_VF1D · 2023-08-17
> > **Requires deeper analysis than in the 1-page addendum**
> >
> > I sincerely appreciate the additional information provided by the authors in their rebuttal.  They have definitely put in major efforts in response to my and other reviewer comments. However the nature of the results provide in the 1-page addendum raise several deeper question as outlined below.
> >
> > Papers along the lines of  [1][2] DropEdge, ShadowGNN etc. attempt to provide some theoretical basis for their oversmoothing claims in terms analyzing variance in embeddings. Eq. (3) in this paper is quite crucial in terms of intuition for preventing oversmoothing, however unlike [1][2] and similar others, results based on Eq. (3) are primarily heuristic and intuitive. Since reduction of oversmoothing is the major claim in this paper, an important question to readers will be whether the TV method adopted in this paper leading to (3) can be analyzed in a similar manner to provide some theoretical basis for the reduction in oversmoothing. The new results presented in the 1-page addendum show a rather remarkable outperformance of node classification of this work on heterophilic datasets compared to ${\bf every}$ other baseline which makes some theoretical basis for oversmoothing even more needed. The new table of results in the 1-page addendum are appreciated but too succinct in terms of experimental details and raise several deep question on how this overperformance is achieved (while not the authors fault since the rebuttal is page limited, but too important since they consider a complete independent area of heterophilic graphs – it would have been great if the authors had given this full treatment with the proper analysis in the expanded version, not just an additional table). An important exceptional result like this should be analyzed comprehensively with explanations, ablation studies, and comparative analysis, the current submission does not provide the means to do so.
> >
> > I have a similar comment for the new graph on traffic flow prediction. What was the input setup, any simplifying assumptions, any specific engineering of meta-parameters etc., what are the limitations of using the proposed method of flow diffusion for such an important problem space. How does the TV method work so well for traffic flow? I think the community will be better served if the paper is resubmitted with the proper treatment of these important applications.

---

> > > ### Author Response · Authors · 2023-08-19
> > > **Response to Reviewer VF1D**
> > >
> > > Dear Reviewer VF1D,
> > >
> > > Thank you for the follow-up comments.
> > >
> > > (1) Existing works such as DropEdge [1] and ShaDowGNN [2] have achieved great success in addressing the over-smoothing issue using graph theory. However, we study the over-smoothing issue from a completely different perspective by formulating graph learning as an ill-posed optimization problem. In general, the energy function is defined to transform the initial graph embeddings (via a graph diffusion process) to the extent that the diffused graph embeddings reach the largest correlation with the outcomes (labels), which is constrained by a pre-defined regularization term (such as $l_2$-norm graph smoothness term). Following the notion of total variation (TV), we introduce a selective gating mechanism that adaptively controls the smoothness based on the learnable threshold (Eq. 3). The step-by-step derivation of Eq. 3 in the framework of variational calculus is detailed in the supplementary S1.1 (please check lines 16-31).  Since our work is built upon the well-studied framework of variational calculus, we paid more attention to interpreting the insights of the selective smoothing mechanism from the TV term (line 191-197 and line 235-243 in the main manuscript), rather than the theoretic basis for the reduction in over-smoothing.
> > >
> > >
> > > (2) The calculus of variations and partial differential equations were extensively employed in the field of image processing several decades ago. It is important to note that we are not reinventing the wheel of existing theoretical concepts. A majority of the theoretical proofs related to TV-based optimization can be located in the textbook "Mathematical Problems in Image Processing: Partial Differential Equations and the Calculus of Variations" by Aubert and Kornprobst, published by Springer. However, what makes our work unique is that our pilot application of the variational calculus framework to graph neural networks. As supported by the state-of-the-art performance (on most graph public datasets) with comparison to existing benchmark GNN models, the integration of these mathematical principles into the realm of graph-based learning could be a new recipe for designing novel application-specific GNNs.
> > >
> > >
> > > (3) We appreciate  reviewer VF1D giving us another chance to explain the important details that we could not put in the 1-page addendum. Actually we were planning to show improvement of each GNN model in Table 2 of the addendum after plug-in the DC layer (marked '+'). The results are summarized as follows.
> > >
> > >            Texas (a) – Wisconsin (b) – Actor (c) – Squirrel (d) – Chameleon (e) – Cornell (f)
> > > GPRGNN+:  0.5762 ($a$) –  0.5375 ($b$)  –  0.2873 ($c$) –  0.2936 ($d$) –  0.4417 ($e$) –  0.4746 ($f$)
> > >
> > > H2GCN+:   0.2881 ($a$)  –  0.2584 ($b$) – 0.1954 ($c$)  –  0.2745 ($d$) –  0.2237 ($e$) –  0.2162 ($f$)
> > >
> > > ShawDowGNN: 0.2140 ($a$) –  0.2367 ($b$) – 0.2112 ($c$) – 0.2010  ($d$)  –  0.2303 ($e$)  – 0.2328 ($f$)
> > >
> > > Note, we only conducted the experiments on 128-layers on involved methods, we will add all the experiments on different network layers to the final version.
> > >
> > > We further conducted an ablation study on GCNII [1] without DC layer, the performance using 128 layers is as follows.
> > >
> > > GCNII:    0.7390 ($a$)   –  0.7575 ($b$)  – 0.3393  ($c$)    –  0.3017  ($d$)  –  0.4513   ($e$) –  0.7119  ($f$)
> > >
> > > [1] Simple and Deep Graph Convolutional Network, ICML 2020
> > >
> > > It is worth noting that we have released the code and data in the anonymous Github. We are committed to showing these new results in the Supplementary of the final version.
> > >
> > > (4) We have a clear neuroscience motivation for characterizing the toxic protein flows from neuroimages. As mounting evidence shows that the spreading of disease pathology underlines the topology of the wiring diagram in the brain (Franzmeier et al. “Functional Brain Architecture is Associated with the Rate of Tau accumulation in Alzheimer’s Disease”, Nature Communication, 2020) to the extent that a large portion of spreading flows occur between strongly interconnected nodes such as hubs. Since the topologically critical nodes (such as hubs) often have high degree of connections, it is reasonable to use TV-based regularization term to avoid vanishing flows along strong links by selectively suppressing the potential over-smoothing of embedding vectors between nodes with dense connections.
> > >
> > > Due to character restrictions, please refer to the next page's response.

---

> > > > ### Author Response · Authors · 2023-08-19
> > > > **Additional Responses to Reviewer VF1D**
> > > >
> > > > (5) In this rebuttal, we follow reviewer VF1D’s suggestions to test our FlowNet on traffic data, in spite of the fact that traffic application is really out of our scope. Since we are new to traffic applications, we used all default settings in terms of data splitting (such as [1]) and the same hyperparameters in the FlowNet on pathology data. In a quick turnaround of rebuttal, we don’t have the bandwidth for the engineering tricks. i.e., each $x_t$ indicates a frame of current traffic status at time step $t$, which is recorded in a graph structured data matrix.
> > > > $x_t \in \mathbb{R}^n$ is a vector of observations for $n$ road segments at time step $t$, where each element contains historical data for an individual road segment.
> > > > We can predict the next time point ($t+1$) traffic flow condition $x_{t+1}$ .
> > > >
> > > > [1] Spatio-Temporal Graph Convolutional Networks: A Deep Learning Framework for Traffic Forecasting, IJCAI, 2018
> > > >
> > > > (6) The exploration on traffic data has demonstrated the effectiveness of new GNN model based on TV. One possible explanation of the promising results might lean to the similar topology structure between brain network and traffic network which both manifest the “small-world” organization pattern (i.e., most nodes are not neighbors of one another, but most nodes can be reached from every other by a small number of hops).
> > > >
> > > > Best,
> > > >
> > > > Authors

---

> ### Comment · Area_Chair_N32h · 2023-08-17
> **Please provide additional feedback**
>
> Hi,
>
> You seem to have the lowest score for this paper. Could you please acknowledge that you have read the rebuttal and let us know if you still have concerns or not? If not, then I would encourage you to increase your score.

---

### Author Rebuttal · Authors · 2023-08-09

We thank all reviewers for their thoughtful feedback.

We are thrilled by the reviewers’ comments which they consider our paper as being original (Reviewer CRUA), creative (Reviewer HWVs), novel yet simple (Reviewer VF1D, sJUp, HWVs), clear (Reviewer sJUp, HWVs), great (Reviewer CRUA), very well  (Reviewer paGB) and well written (Reviewer VF1D).

We are glad they found our study provided solid mathematical support  (Reviewer CRUA) and the rationality of the proposed theory (Reviewer HWVs), as well as our analyses are thorough (Reviewer paGB), well-defined (Reviewer HWVs) and the great and superior experimental results (Reviewers paGB, CRUA, sJUp) strongly support the claims.

We agree with the Reviewer HWVs who recognizes the detailed knowledge of the EL equation is not familiar to every graph neural network researcher. We were constrained by space. Therefore, we will add more details and relevant work in the Supplementary of the final version.

One primary concern is limited evaluations of the experiments on comparison methods and datasets (Reviewer VF1D). We have added three comparison methods (Reviewer VF1D mentioned), six benchmark datasets of heterophilic graphs and two benchmark data of traffic flow. All the results are provided in the 1-page PDF.

$ \bf Table 1$ (for Reviewer VF1D): The performance of different data splitting schemas.

$ \bf Table 2$ (for Reviewer VF1D): Node classification results on heterophilic graphs (including six datasets) for four methods.

$ \bf Table 3$ (for Reviewers VF1D, CRUA, paGB, sJUp): The running time on different methods.

$ \bf Fig 1$ (for Reviewers VF1D): Traffic flow prediction accuracy in terms of mean absolute error (MAE) in PEMS-BAY and METR-LA benchmark datasets.

$ \bf Fig 2$ (for Reviewers HWVs): The t-SNE visualization of node feature representations.

$ \bf Fig 3$ (for Reviewers HWVs): The evolution of $l_2$-norm graph smoothness term and $l_1$-norm TV term as the number of GNN layers increases.

We have answered all the specific questions for every reviewer as below and will incorporate all feedback in the final version.

---

### Comment · Area_Chair_N32h · 2023-08-17
**Comment by the AC**

Dear Authors,

Thank you for your detailed answers and the effort that you put in your rebuttal.

Could the reviewers who haven't responded to the authors please do this as soon as possible.

---

### Decision · Program_Chairs · 2023-09-21

**Decision:**

Accept (poster)

**Comment:**

Most reviewers support the paper. So I recommend acceptance as a poster.